# Arctic Regional Methane Fluxes by Ecotope as Derived Using Eddy Covariance from a Low-Flying Aircraft

David S. Sayres[1], Ronald Dobosy[2,3], Claire Healy[4], Edward Dumas[2,3], John Kochendorfer[2], Jason Munster[1], Jordan Wilkerson[5], Bruce Baker[2], and James G. Anderson[1,4,5]

[1]Paulson School of Engineering and Applied Sciences, Harvard University, 12 Oxford Street, Cambridge, MA 02138
[2]Atmospheric Turbulence and Diffusion Division, NOAA/ARL, Oak Ridge, TN 37830
[3]Oak Ridge Associated Universities (ORAU), Oak Ridge, TN 37830
[4]Department of Earth and Planetary Sciences, Harvard University, 12 Oxford Street, Cambridge, MA 02138
[5]Department of Chemistry and Chemical Biology, Harvard University, 12 Oxford Street, Cambridge, MA 02138

*Correspondence to:* David Sayres (sayres@huarp.harvard.edu)

**Abstract.** The Arctic terrestrial and subsea permafrost region contains approximately 30% of the global carbon stock, and therefore understanding Arctic methane emissions and how they might change with a changing climate is important for quantifying the global methane budget and understanding its growth in the atmosphere. Here we present measurements from a new *in situ* flux observation system designed for use on a small, low-flying aircraft that was deployed over the North Slope of Alaska during August, 2013. The system combines a small methane instrument based on Integrated Cavity Output Spectroscopy (ICOS) with an air turbulence probe to calculate methane fluxes based on eddy covariance. We group surface fluxes by land class using a map based on LandSat Thematic Mapper (TM) data having 30-meter resolution. We find that wet sedge areas dominate the methane fluxes with a mean flux of 2.1 $\mu g \cdot m^{-2} \cdot s^{-1}$ during the first part of August, with methane emissions from the Sagavanirktok River being the second highest at almost 1 $\mu g \cdot m^{-2} \cdot s^{-1}$. During the second half of August, after soil temperatures had cooled by 7 °C, methane emissions fell to between 0 and 0.5 $\mu g \cdot m^{-2} \cdot s^{-1}$ for all areas measured. We compare the aircraft measurements with an eddy covariance flux tower located in a wet sedge area and show that the two measurements agree quantitatively when the footprints of both overlap. However, fluxes from sedge vary at times by a factor of two or more even within a few kilometers of the tower demonstrating the importance of making regional measurements to map out methane emission spatial heterogeneity. Aircraft measurements of surface flux can play an important role in bridging the gap between ground-based measurements and regional measurements from remote sensing instruments and models.

## 1 Introduction

Methane is the third most important greenhouse gas after water vapor and carbon dioxide, and its concentration in the atmosphere has increased from a pre-industrial value of 0.7 parts per million by volume (ppmv) to its current value of approximately 1.85 ppmv. Methane sources are varied, with major contributors being anthropogenic (including fossil and agricultural) as well as natural. Often multiple sources occur in the same vicinity, for example emissions from gas wells collocated with agricultural fields or pasture for grazing livestock.

In the past few years there have been increased efforts to understand how methane emissions, as well as carbon dioxide, might change from the Arctic region in response to warmer temperatures (Yvon-Durocher et al., 2014; Sturtevant et al., 2012; Sturtevant and Oechel, 2013; Walter et al., 2007b, and references therein). For example, temperatures in the Alaskan North Slope have increased 0.6 °C per decade for the last 30 years. Likewise, in that same time period the minimum extent of Arctic

sea ice at the end of the summer has decreased from 8 million $km^2$ to 5 million $km^2$. Until this past century late-summer sea ice extent was 10 million ± 1 million $km^2$ over the past 1500 years (Kinnard et al., 2011). Global methane concentrations have also varied during this time period, with atmospheric increases slowing down in the 1990s, leveling off in the early part of the 21st century and then increasing again since 2007 with concentrations reaching 1.8 ppmv in 2010 based on several surface based observation networks (Kirschke et al., 2013). It has been postulated that the increase could be from Arctic wetlands

(Koven et al., 2011; Walter et al., 2007b).

A brief look at the carbon stock in the Arctic reveals why it has garnered so much attention. The Arctic permafrost region contains between 1330 and 1580 Pg of carbon in the tundra surface layer (0-3 meters depth), Yedoma deposits, and rivers. An additional quantity is contained in deeper deposits and subsea permafrost (Tarnocai et al., 2009). Arctic carbon stock represents about a third of the total global surface carbon pool and increases to 50% when accounting for the deeper soils (Schuur et al.,

2015). As the climate continues to warm, this carbon is vulnerable to thaw and decomposition by microbes, potentially leading to large increases in methane and carbon dioxide emissions. Methane from anaerobic reduction of organic carbon stocks in permafrost is particularly important, having a warming potential more than twenty times that of carbon dioxide on a 100-year time scale and greater yet over shorter time periods (Boucher et al., 2009). The correlation between a warming Arctic and the release of methane and carbon dioxide from northern wetlands and ocean clathrates is strongly evident in the paleoclimate

record (Zachos et al., 2008; Whiticar and Schaefer, 2007). This relation is also seen 1) in current observations of methane release from thermokarst lakes formed from melting Arctic permafrost each spring and summer (Sepulveda-Jauregui et al., 2015; Walter et al., 2007b; Bastviken et al., 2004; Casper et al., 2000), 2) in ebullition from deep sea sediments (Shakhova et al., 2014; Reagan et al., 2011; Damm et al., 2010), and 3) from airborne campaigns (Wofsy, 2011; Chang et al., 2014).

The North Slope of Alaska is covered by several different land classes though dominated by permafrost. Access to the interior

normally requires aircraft, except along the Dalton Highway (Rt. 11) from Fairbanks to Prudhoe Bay. The lack of infrastructure, especially roads, makes continuous ground-based measurements difficult except near the major settlements. This sparsity of data increases the uncertainty in regional bottom-up estimates of carbon flux. At the same time top-down estimates based on inversion modeling from measured concentration profiles rely on knowledge of flux sources on the ground to determine which sources are dominating the emissions in areas like the North Slope with its multitude of broad-scale emitters and point sources.

A scale gap exists between process-level studies on the ground and large-scale regional estimates from remotely sensed data or inversion-model results. Airborne measurements, especially from low-flying aircraft, have the potential to bridge this gap. Flux measurements from low-flying aircraft coordinated with surface measurements promote extension of the detailed surface-flux measurements to the larger regional scale by mapping the heterogeneity in the fluxes over these larger areas.

Eddy covariance is a direct way to determine *in situ* the exchange (flux) of mass, momentum, and energy between the

atmosphere and the surface. Turbulent wind and concentration are measured at high sample rate, and their covariance yields

the flux. With stationary instruments the wind and concentration measurements can be routinely obtained, and eddy covariance from fixed sites is widely represented in the literature as a way of obtaining the flux of a quantity between the surface and atmosphere. Obtaining eddy covariance measurements from a moving aircraft presents some unique challenges including accurately measuring turbulent wind velocity relative to the ground and measuring concentration at a sufficiently high data rate. Furthermore, if the flux from the aircraft is to be a good proxy for a measurement taken at the surface, it needs to be sampled close to the ground. The appropriate distance varies depending on boundary layer height, turbulence, and the footprint size of interest. Several groups have successfully measured carbon dioxide and heat flux from low flying aircraft in the Arctic (Zulueta et al., 2011; Oechel et al., 2000, 1998; Gioli et al., 2004), Europe (Bange et al., 2007; Vellinga et al., 2010; Hutjes et al., 2010; Gioli et al., 2006), Asia (Metzger et al., 2013), and continental US (Kirby et al., 2008; LeMone et al., 2003; Avissar et al., 2009).

Here we present methane fluxes taken during the summer of 2013 in the North Slope of Alaska and use the data to explore several questions. For example, how representative are towers' footprints of other instances of the remotely determined class of land cover in which they were placed? In principle a stationary site can measure all manner of properties and state variables in the soil, the vegetation, and the air within and above the canopy. Much can be learned about the bacteria, soil chemistry, canopy storage, and other quantities relevant to the exchange of mass, momentum, and energy with the surface. But all of this is known only at the one site. How representative is that site of other locations that to remote sensors appear similar? Are there land-cover types that are particularly indicative of emission of a given trace gas? Can the class so identified be used as a quantitative predictor of a particular type of soil chemistry? This is relevant in assessing the regional methane emission from remote sensing. Methane in particular has a fairly complex chemistry in the soil involving state quantities such as the (sub-canopy) soil temperature and the height of the water table. These are measurable only in situ so that having a proxy indicator such as vegetation cover would be valuable. Interval quantities[1] sensible remotely, such as NDVI, air temperature, and other vegetative indexes that correlate with carbon dioxide do not correlate with methane (Olefeldt et al., 2013). Vegetation classifications determined remotely, however, have been shown in other regions to be useful for estimating regional methane emissions (eg. Schneider et al., 2009) in upscaling from ground measurements.

Aircraft, though more limited in what they can measure than fixed sites, are very mobile providing the opportunity to sample many instances of the same remotely sensed class over the landscape. From this multi-instance sample one can assess how representative the single fixed site is. One can also assess the strength of the variability within the given land-surface class for later investigation from the surface. In remote parts of the earth, in particular, if surfaces of recognizably similar character (class) can be shown to have comparable emissions properties, considerable effort can be saved over a surface-based survey. Alternatively, large variation within a class that is not currently well predicted by some remotely measurable interval quantity will be seen as requiring additional effort for in-situ measurements to find an effective monitoring program for methane emission from that surface class.

---

[1]An interval quantity such as temperature can take an ordered range of values the length of which has meaning, as opposed to a set of categories such as surface classes having no notion of order or length.

## 2 Methods

To measure methane emissions over large areas of the North Slope, the Flux Observations of Carbon from an Airborne Laboratory (FOCAL) system was flown during August 2013 out of Deadhorse Airport, Prudhoe Bay, AK. FOCAL, pictured in Fig. 1 flying near the NOAA Atmospheric Turbulence and Diffusion Division (NOAA/ATDD) flux tower, consisted of three

main parts: the aircraft, a Diamond DA-42 from Aurora Flight Sciences, a turbulence probe, the Best Airborne Turbulence (BAT) Probe from NOAA/ATDD, and a fast methane and water instrument from the Anderson Group at Harvard University. Data presented in the results section were obtained during six flights between August 13 and August 28 (Fig. 2 and Table 1). During three of these flights the aircraft made repeated passes near the NOAA/ATDD tower that was set up for comparisons. The other three flights were flown as grid patterns over large regional areas ($\sim$50x50 km$^2$) to better sample the heterogeneity of

different land types over a large region. These flights consisted of both profiles from the bottom of the boundary layer ($\sim$5-10 m) up to $\sim$1500 m altitude and long transects ($\sim$50 km) at low altitudes (<25 m) that are used to access surface flux using eddy covariance.

### 2.1 FOCAL instrumentation

Fluxes of trace gases are covariances between turbulent winds and fluctuations in gas concentration. The airborne methane flux

calculations rely on having fast measurements of both turbulent wind velocity and dry-air mixing ratio, with the two quantities being coordinated in time and space to within an error much smaller than the measurement interval.

To measure turbulent wind, temperature, and pressure NOAA/ATDD developed the BAT probe in the 1990s as a pioneering low-cost solution for mobile atmospheric turbulence measurements (Crawford et al., 1996, 1993; Crawford and Dobosy, 1992). The BAT probe consists of a hemisphere, 15.5-cm in diameter, with nine pressure ports located at selected positions on the

probe head. The vertical and horizontal pairs of ports measure the differential pressure between them to calculate the angle of attack and side slip, respectively. Static pressure is taken from the average of the pressures measured at the four diagonal pressure ports corrected for nonzero attack and sideslip angles. Dynamic pressure is measured from the difference between the pressure measured at the center hole and the static pressure, again adjusted for nonzero values of the angles of attack and sideslip. These pressure measurements are combined with a known model for flow over a hemisphere to determine three

dimensional wind direction and speed relative to the probe. The velcity of the probe relative to the ground is measured by three interconnected instrument systems: a GPS/INS system located near the center of gravity (CG) of the aircraft, accelerometers located in the probe, and two additional GPS antennas, one on the BAT probe and the other atop the the main cabin(Crawford and Dobosy, 1997, 1992). The BAT probe digitizes samples at 1600 s$^{-1}$, applies a low-pass filter to suppress aliasing, and subsampling at 50 s$^{-1}$ . The wind measurements are synchronized with the 50 s$^{-1}$ signal from the GPS/INS system.

Before assembling the FOCAL system, the BAT probe was characterized in a wind tunnel (Dobosy et al., 2013). A similar BAT probe was also tested in flight on a different aircraft (Vellinga et al., 2013, hereafter V2013). After the FOCAL system was assembled, similar calibration maneuvers were flown in preparation for and during the Alaska campaign. As part of a calibration flight on the evening of August 27 in Alaska, we performed the yaw maneuver described by V2013 and obtained

a residual contamination less than 10%, as described there. A pitch maneuver described by V2013 was performed resulting in contamination of 10% for the high-frequency pitching (1.6 s period), which was the best executed of the pitch test's three parts and is the severest test.

The methane instrument draws air from an inlet located 8 cm aft of the BAT probe turbulence measurements. Flow of
air through the axis is controlled by a dry scroll pump located in the back of the aircraft. Air from the inlet passes through 1.25 cm diameter tubes into the nose and forward luggage bay sections of the aircraft. The pressure of the air is controlled by a proportional solenoid valve and a pressure control board that uses pressure measured at the detection axis to feed back on the valve orifice position. The actual detection axis is located in the port-side forward luggage bay. The methane instrument uses Integrated Cavity Output Spectroscopy (ICOS) to measure $CH_4$, $H_2O$ and $N_2O$ (Witinski et al., 2011). The ICOS instrument
uses a high-finesse optical cavity composed of two high-reflectivity mirrors (R = 0.9996) to trap laser light for a period on the order of 2 μs producing effective path lengths $10^3$ times the mirror separation. For the fast methane sensor used in this deployment a small ICOS cell (25 cm in length; mirrors 5 cm in diameter) was built that combines the sensitivity and stability of ICOS with a small sample volume to attain high flush rates ($17 \, \text{s}^{-1}$), permitting a sample rate of $10 \, \text{s}^{-1}$. Using the wavelength region around $1292 \, \text{cm}^{-1}$ (7.74 μm), measurements of methane achieved a precision of 7 ppbv ($1 - \sigma$, $1 - \text{s}$). Due to the high
variability of water in the troposphere, water vapor measurements are required with any trace gas measurements in order to quantify dilution effects caused by changes in water vapor content as well as changes to spectroscopic line broadening (Webb et al., 1980; Gu et al., 2012). Well defined absorption features of water vapor and its isotopologues as well as nitrous oxide are obtained in the same sweep of the laser, therefore the same instrument provides simultaneous measurements of nitrous oxide and water vapor along with methane. This technique provides an extremely high signal-to-noise ratio as well as a robust
measurement in flight and has been the basis for several ICOS flight instruments built by this group (Witinski et al., 2011; Sayres et al., 2009; Engel et al., 2006; Paul et al., 2001). Periodic calibration in flight using calibrated gas cylinders tracks the drift over the course of the flight and from flight to flight.

To match the vertical wind's sample rate, gas series are interpolated to $50 \, \text{s}^{-1}$ using cubic splines. On some of the flights a buffer overflow problem (since corrected) caused sample loss leaving an irregular time series of samples between $3 \, \text{s}^{-1}$ and
$4 \, \text{s}^{-1}$. The irregularity was readily handled by the interpolation to produce a signal, implicitly low-pass filtered with a stop band above about 1.5 Hz, down from the full 5 Hz ($10 \, \text{s}^{-1}$). Plots of spectra and cospectra of the data streams of the vertical wind and of the trace gases' dry-air mixing ratios were prepared and are presented by Dobosy et al. (2017). To assess the potential loss of flux due to the lost samples, the full $10 \, \text{s}^{-1}$ gas-data stream available from flight 25.18:00 was subsampled in two modes. One subsample was evenly spaced at $3 \, \text{s}^{-1}$; the other more randomly spaced between $3 \, \text{s}^{-1}$ and $4 \, \text{s}^{-1}$, representing flight 13.09:30.
These were interpolated by cubic spline, which does not appreciably add higher-frequency components to the gas-data streams above the (effective) Nyquist frequency of the original signal (5 Hz for 25.18:00 and about 1.5 Hz for 13.09:30). The test indicated a loss of about 10% of the flux for either subsample. This was considered acceptable for the present study.

High-frequency spectral corrections were not used in computing the fluxes presented here. The resulting loss is less than 10%, as confirmed in the implicit filtering test above. A second test, differing only in the filter used provides further confir-

mation. A four-pole Butterworth low-pass filter is applied forward and backward to cancel the phase shift. Four cases were simulated using data from flights 25.18:00 and 13.09:30.

1. Filter the gas series to a 2 Hz cutoff (half-power). This first reduction has almost no effect on 13.09:30 since it is already filtered as discussed above.

2. Filter the gas series to 2 Hz; also filter the vertical wind to 2 Hz. This had a small additional effect. The flux published in this paper used the full-frequency wind data.

3. Same as 1, but filter gases to 1 Hz.

4. Same as 2, but filter both gases and vertical wind to 1 Hz.

All filters were implemented on the merged wind- and gas-data series at $50\ \mathrm{s}^{-1}$. The simulation provides an upper bound on the loss of flux above the 5-Hz cutoff frequency of the full gas-data streams. Cutting the effective Nyquist frequency down to 2 Hz and then further to 1 Hz cuts more and more deeply into spectral ranges having increasingly significant contribution to the flux. This is reflected in the results: 10% loss at 2-Hz Nyquist frequency and 28% loss at 1 Hz. The results indicate that the fraction of flux lost from frequencies higher than 5 Hz is less than 10%. Future work will, however, include exploration of these estimation techniques.

Finally, to evaluate the dependence of the measured methane flux on the height above the ground, a regression of the 3-km running flux (see Sec. 2.3.1) from flight 13.09:30 was run against flight altitudes ranging from 5 to 45 m. A quadratic regression was required yielding significant positive slope but significant negative curvature. The regression line reached a maximum at an intermediate point before the maximum height above ground. More importantly, the regression explained only 10% of the variance.

There were two other small instruments that augmented FOCAL's capabilities: a radar altimeter, for height above ground which is essential for accurate footprint calculations, and a visible-light camera, which provided a visual record of the terrain directly under the aircraft to check the accuracy of the remotely sensed products used for primary landscape classification. The Aurora Flight Sciences' version of the DA-42, named the Centaur, is a twin-engine aircraft having several characteristics that make it an ideal platform for the work discussed here. The Centaur's twin-engine configuration leaves the entire center fuselage available for instrumentation and sampling. The aircraft is electrically and structurally well-adapted for carrying a sophisticated scientific payload, having ample spare power from its two alternators and ideally located hard points for the probe and the spectroscopic equipment.

## 2.2 Turbulence measurements

Eddy covariance is a direct way to determine the exchange of mass (e.g., trace gases), momentum, and energy between the atmosphere and the surface. In principle for a gas, the covariance between the turbulent fluctuating gas concentration and the turbulent vertical wind component determines the flux. Since the flux thus obtained is assumed to represent the exchange at the surface, the airplane is flown as low as is safely possible, typically below 30 m (Mahrt, 1998). Flux measurements from fixed

surface sites, important complements to the airborne measurements, provide extended temporal coverage at selected locations as well as validation of the airborne flux measurements.

The mass flux of a minor gas constituent in air, such as methane, is calculated following Webb et al. (1980); Gu et al. (2012). Let $\rho_a$ be the the partial density of air apart from water vapor and $w$ be the vertical wind velocity. Then $\rho_a w$ is the dry-air mass flux, which is expanded into base state and turbulent departure with the base state represented by an overbar and the departure by a prime:

$$\rho_a w = \overline{\rho_a w} + (\rho_a w)'. \tag{1}$$

Since dry air is not exchanged with the surface, $\overline{\rho_a w} = 0$. The flux of a gas is then the covariance of the turbulent dry-air mixing ratio $c'$ with the turbulent dry-air mass flux $(\rho_a w)'$:

$$F = \overline{(\rho_a w)' c'}. \tag{2}$$

Unlike from a stationary tower, measuring the turbulent vertical wind component from an airplane requires finding the small (vector) sum of the airspeed and the ground speed, two large, nearly canceling vectors. Since both vectors fluctuate rapidly and independently, many independent measurements must be made with precise synchrony at high accuracy and sample rate. Since turbulent fluctuations can be less than $0.1 \mathrm{\ m \cdot s^{-1}}$, the two large velocities must each be accurate within $0.1 \mathrm{\ m \cdot s^{-1}}$. Four samples define the minimum effectively resolvable turbulent eddy size, about 5 m at 50 samples per second and $60 \mathrm{\ m \cdot s^{-1}}$.

## 2.3 Methane Flux Measurements

### 2.3.1 Running Flux Method

The running flux method (RFM) is commonly used in the space/time domain for eddy covariance analysis of airborne fluxes (e.g. LeMone et al., 2003). The RFM calculates the mean flux over a contiguous integration length (e.g., 3 km). As opposed to a stationary tower, which averages in time, the aircraft is moving over the landscape, so that fluxes are more appropriately averages over distance. Here we use the same notation as Crawford et al. (1993)

$$F = \frac{\sum_{k=1}^{N} (\rho_d w)'_k c'_k V_k}{\sum_{k=1}^{N} V_k} \tag{3}$$

where $\rho_a$, $w'$, and $c'$ are defined as in Eq. (2) and $V$ is the airspeed of the aircraft. The sum is over $N$ consecutive samples, and the denominator is the spatial averaging length. For the analysis presented here we use a 3 km window that is moved by 1 km increments so that, unlike the normal practice with tower data, there is overlap between adjacent calculated fluxes to provide somewhat finer spatial localization. The RFM quantitatively describes the relation between measured flux and underlying surface features of scales comparable to the averaging length or larger. This method works well as shown by LeMone et al. (2003) who found a 4 km moving average on the US Great Plains to be an appropriate compromise between uncertainty in flux estimation and resolution of landscape-scale heterogeneity. In the Arctic in 2013, the much smaller mixed layer depth gave rise to smaller turbulence scales. Ogive analysis of the frequency distribution showed 3 km to suffice as the integration distance

(Berger et al., 2001). However, heterogeneity in the resulting flux estimates was large. Repeated flight segments gave variable results likely due to changes in winds and sampling footprints and to the integration lengths being longer than the scale of the underlying surface features. Nevertheless, there was good agreement between methane fluxes calculated by the RFM using 3 km integration centered near the tower location and fluxes computed directly from the tower measurements (see sec. 3.1).

Using the RFM over the small-scale heterogeneity of the North Slope's surface features, however, limits the ability to isolate the flux contributions from individual surface classes.

### 2.3.2 Flux Fragment Method

The Flux Fragment Method (FFM) was conceived to assess the homogeneity in properties of a remotely determined land class over multiple instances occurring in patches on the landscape. Often such patches are too small for a traditional RFM (Kirby

et al., 2008). The FFM, while based on the same statistical foundation as eddy covariance, uses a conditional sampling scheme whereby the flux, of methane for example, is compiled from many $\tau$-second 'fragments' $f_i$ of methane flux along a transect, each given by

$$f_i = \delta t \sum_{k=1}^{n\tau} [(\rho_d w)'_k c'_k V_k]_i \qquad (4)$$

$$L_i = \delta t \sum_{k=1}^{n\tau} [V_k]_i. \qquad (5)$$

Here $n$ is the number of samples per second, $\delta t$ is the sample interval, and everything else is defined as in Eq. (3) except that instead of summing over a large distance, such as 3 km, the sum is only over a few samples. Note, however, that the departure quantities used to form the fragments are relative to the same base state as in Eq. (3), a base-state of 3-km scale or more, determined by ogive analysis (Foken, 2008) to be an upper limit for the turbulence present at the time of measurement. The fragments therefore contain information on all scales from the Nyquist wavelength of the sample rate up to the 3-km

scale of the spectral gap determined from the ogive analysis. Yet, the air packets quantified by the fragments are also short enough to have likely interacted with a single class of surface. So long as any significant secondary circulations are accounted in the base-state, the turbulent atmosphere on all its scales can be postulated to repeat over the landscape in a fairly random fashion. A contiguous sample (i.e, without gaps) should not therefore be required. The sample only need be sufficiently large to include multiple instances of boundary-layer structures at each scale. An aircraft moving at airspeed $60\,\mathrm{m\cdot s^{-1}}$ covers 216

km in an hour encountering 72 instances of 3-km turbulence structure. A sufficiently prevalent class of land surface, whether found in large or small patches is very likely to provide a sufficient sample. Samples which are too short can be discovered in confidence intervals developed by bootstrap resampling as was done by Kirby et al. (2008). A more sophisticated bootstrap procedure developed in conjunction with analysis of these 2013 data by Dobosy et al. (2017) follows Mudelsee (2010).

In the data presented here the fragments are 1-s sums ($\tau = 1$ s) of approximately 60 m length. The fragments, labeled $f_i$, do

not constitute a Reynolds average individually. That is, an individual fragment, though containing all turbulent scales, is only a short grab sample. Fragments provide a meaningful flux estimate only in aggregate. They can be grouped, for example, by surface class, determined from footprint estimation (Fig. 3). Fluxes are calculated only for those surface-class groups whose

total length is greater than 3 km. The sum over each group divided by the cumulative length of all fragments in the group provides the mean flux from the associated surface class as given by

$$F_S = \frac{\sum_{i \in S} f_i}{\sum_{i \in S} L_i} \tag{6}$$

The FFM is most appropriate in a region that is heterogeneous on small scales (100 m to 3 km), but relatively homogeneous on large scales such that many instances of the surface class, or other classification used to group the fragments, are sampled during the flight (See Kirby et al. (2008) for the full description of the method). Initial assessments of the data presented here indicate that the FFM is well suited for application to the North Slope, where Arctic tundra is interspersed with thermokarst lakes, bogs, fens and bare ground. First, land-cover data are classified using a current land-cover image at 100 m resolution or better (e.g. LandSat). We use this to establish transects flown at altitudes typically 10 m to 30 m above ground; as low as safely possible. These are flown repeatedly and coordinated with eddy-covariance towers for validation and temporal continuity. The base state is then defined, representing in principle the deterministic (non-turbulent) mesoscale component of the flow. Flux fragments are calculated using 1 s sums of squares and cross products of departures from the base state. Finally, a footprint model is applied to estimate the level of influence of each surface type on each fragment. See sec. 3.2 for examples of how FFM is used to interpret these data.

For the questions to be addressed in this paper the footprint model provides a measure of a fragment's membership in the fuzzy set (Nguyen and Walker, 2000) associated with each surface type, treated as a categorical variable. Fragments having a sufficient level of membership for a particular surface class are assigned to that class. A membership level above 0.5 restricts all fragments to a maximum of one class. Fragments can thus be grouped into sets in which all members have a measure greater than the prespecified level that they came from the same surface type.

We use the parameterization scheme developed by Kljun et al. (2004) from a set of runs of a backward Lagrangian model (Kljun et al., 2002) for a range of heights, stability measures and other turbulence quantities that are measured from the aircraft. The required turbulence quantities are computed from averages taken over the length of each flight leg, where the flight leg is defined as the straight segment, between turns, over which the collected data are used. The more recent two-dimensional version (Kljun et al., 2015), which was considered too computationally intensive to be included in the present study, was not considered necessary because of the footprint's current restricted use as a membership criterion to assign a selected subset of fragments to the surface categories. The degree of overlap was assessed, however, for future reference. Using the measurements from the convective daytime case 13.09:30 the two-dimensional model yielded a footprint with a full width of about 250 m $(\pm 1 - \sigma_y)$ at the location of maximum crosswind-integrated probability, 93 m upwind of the sensors. Since the probabilities are weighted towards the middle of the footprint and the land classes tend to be homogeneous on the order of at least 300 m, using the one-dimensional version of the model is acceptable given our focus on categorical classification and our strict membership criterion (85%). With interval quantities the weighted distribution of sources over the full two-dimensional footprints will be required.

The flux estimate for each land surface type is the sum of the fragments in the associated group divided by their accumulated length. The number of fragments necessary to provide a robust result can be determined by bootstrap resampling (Kirby et al., 2008). For the data presented here 3 km or ∼50 fragments suffice.

The questions to be answered by the FFM, using a fuzzy-logic approach (Nguyen and Walker, 2000) to assign surface classes
to fragments and then to conditionally sample them based on those classes include:

1. What is the mean flux over all measured instances of each surface class?

2. What surface classes dominate the methane emission, and by how much?

3. How much does the flux over each class vary? Is there a spatial pattern to the variation? The variability will come both from the prevailing atmospheric environment and the heterogeneity of the emission within the same class.

4. How well does a particular instance represent all similar instances over the landscape?

## 2.4   Land Surface Classification

The land surface on the North Slope can be divided into different classes based on dominant plant species, topography, soil content, and soil moisture. The North Slope Science Initiative (NSSI) has identified 24 classes using Landsat Thematic Mapper (TM) 30 meter resolution land cover maps in conjunction with field surveys (Initiative, 2013). These classifications are plausible
proxies for properties that have been shown to be primary drivers of methane production and emission, including water table height, soil temperature, and emission pathways such as sedge roots. The areas sampled by FOCAL (Fig. 2) were covered by patches of wet sedge, mesic sedge - dwarf shrub, fresh water marsh, tussock tundra, and open water. Open water is visible from the air, and includes lakes of various sizes and origin along with rivers. Coastal waters, however, are excluded for this analysis. By definition in the tussock-tundra land class, shrubs more than 20 cm tall occupy less than 25% of the surface, and
tussocks occupy more than 35%. The sites are cold, poorly drained and underlain by moderately moist (mesic) to wet mineral soils with silty to sandy texture and a shallow surface organic layer surrounding the tussocks. Wet sedge sites are defined as those with sedge species accounting for more than 25% of the cover and Sphagnum for less than 25%. Soils range from acidic to non-acidic, are saturated during the summer, and typically have an organic layer over silt or sand. Mesic sedge - dwarf shrub has shrubs less than 25 cm tall covering more than 25% of the area, and sedge cover is also more than 25%. Soil surface is
generally mesic, but sometimes wet and is calcareous to acidic. The fresh water marshes (FWM) are semi-permanently flooded, but some have seasonal flooding, and the water depth typically exceeds 10 cm. Soils are muck or mineral, and the water can be nutrient-rich.

We use land types defined by a remote measurement, as opposed to soil properties such as moisture, organic carbon content, temperature, etc. because the remotely based definition is more appropriate to comparing to larger regional scale models and
satellites. Thus the land type here is usually a proxy for general classifications of areas with different soil moisture and other properties which are likely the primary drivers of differences in methane emissions. Certain plants such as sedge, however,

have been shown to act as conduits directly facilitating methane release from the soil to the atmosphere through the plants' vascular system (Olefeldt et al., 2013).

In order to distinguish the contribution to the total methane flux from individual land types and to assess the variability across ecotopes, the data are filtered to only include flux fragments having a membership score of at least 85%, determined by integrating the length of the footprint's centerline weighted by the crosswind-integrated probability density that the flux came from a single surface class. Increasing this threshold increases the link between the calculated flux and a single land class, but reduces the number of footprints available for the analysis thus widening the confidence interval. Varying the threshold between 80% and 95% produces only a small effect on the quantification of flux from each land class. We find that 85% is a good compromise between singling out individual land classes while still retaining a sufficient dataset. For the flight speed of the Centaur at low altitude and wind conditions during the flights, the length of the footprint contributing more than 90% of the flux for each 60-m fragement, varied between 100 and 800 m. The above filter eliminates about a third to half of the flux fragments from each flight. Of those, we limit the land classes to those where the total number of flux fragments is more than 50 fragments or an equivalent distance of 3 km. The flux fragments are summed and then divided by the total integration length for each land cover type (Fig. 7).

## 2.5 Tower Measurements

Starting a few weeks before the flight campaign and throughout the month of August, a small portable flux tower was installed at 70.08545° North latitude, 148.57016° West longitude, just south of Prudhoe Bay off the Dalton Highway. During that time the tower recorded $CO_2$ flux, $CH_4$ flux, latent heat flux, sensible heat flux, air temperature, and incoming radiation. Soil temperature probes were used to record soil temperature at 2-cm, 5-cm, 10-cm, and 20-cm depth at three different locations around the tower. The tower was situated in an area dominated by sedge grass, and the surrounding area's water table was frequently near the surface such that the surroundings were puddled and muddy, especially in late August 2013. On the NSSI map the area is labeled as wet sedge. Low light and limited convective mixing are common on the North Slope of Alaska, and data collected in very weak wind do not provide reliable eddy-covariance flux measurements. Consequently, data were removed from the final set when the standard deviation of the vertical wind speed was less than $0.1 \, \mathrm{m \cdot s^{-1}}$.

## 3 Results and discussion

### 3.1 Comparison between aircraft and tower fluxes

On August 13, 25, and 27 the FOCAL aircraft flew repeated passes over a constant northeast/southwest track near the tower affording direct comparison between eddy-covariance methane flux measured from the tower and from the moving aircraft in both RFM and FFM modes (Fig. 2). The flight track was displaced north or south depending on the forecast wind direction so that the aircraft footprint could pass over the tower footprint. For the northerly winds on August 13 and 25, the flight track was displaced south of the tower. For the easterly winds of August 27 the track passed north of the tower.

Two factors, diurnal and seasonal, influenced the fluxes at the tower site (Fig. 4). The flight 13.09:30 on August 13 (DOY-225) was in the daytime earlier in August, when the turbulence was stronger and the soil temperatures at 10-cm depth were 10-14 $^{\circ}$C. The 30-minute-mean methane fluxes at the tower ranged from 1 to 2.5 $\mu g \cdot m^{-2} \cdot s^{-1}$. The flights 25.18:00 and 27.19:00 on August 25 and 27 (DOY-237 and -239) were in the evening and later in August with weaker turbulence and lower soil temperatures of 3-6 $^{\circ}$C at 10 cm depth. Most 30-minute mean methane fluxes ranged from 0.5 to 1.3 $\mu g \cdot m^{-2} \cdot s^{-1}$. The observed variation with soil temperature is consistent with previous studies (e.g. Yvon-Durocher et al., 2014). Aircraft methane fluxes were compared with the tower in two modes: as local RFM, the mean over all transects of a flight of the 3-km flux blocks downwind of and centered nearest to the tower, and as FFM, the mean of the fragments from wet sedge gathered from the whole 50-km transect and the whole flight.

Agreement between the aircraft and tower by local RFM (orange circle), near the tower but not differentiated by surface class, is within the confidence intervals of the data from 13.09:30 and 25.18:00. For 27.19:00 the aircraft measured significantly lower methane flux by local RFM than the tower. By FFM from wet sedge (red line), the same surface class as the tower but not local to it, the methane flux from 13.09:30 agrees very well with the magnitude of the flux measured on August 13 at the tower. However, for 25.18:00 the FFM flux from wet sedge is significantly lower than the August 25 tower measurement. It is likewise for 27.19:00, though the FFM flux over wet sedge is closer to the corresponding tower flux on August 27 than is the flux calculated by the local RFM.

The results from the three near-tower flights represent three different situations. On August 27 (flight 27.19:00), the footprint of the airborne measurement (Fig. 5, bottom panel) differed from that of the tower. On that flight the footprint analysis indicates the highest probability of influence on the RFM flux (red to maroon contours in Fig. 5) to be over open water, not wet sedge, for at least half the range (the 3-km length centered nearest to the tower in the downwind direction). Lakes have been shown to be sporadic hot spots of methane ebullition, but at least at the time of flight these lakes showed very low methane emission. On August 27, the sedge, which makes up more than twice as much of the transect as the lakes, is visible to the FFM, but not to the local RFM near the tower. Also, the turbulence on August 27.19:00 was weak, with $\sigma_w \sim 0.15 \, m \cdot s^{-1}$. This is a case where some signal may have been lost due to insufficient sample rate for the altitude, or perhaps because the measurement was made above the shallow layer of "constant" flux. This is a tradeoff that plagues evening and morning flights. Notable about flight 27.19:00 is its demonstration of the need for, and difficulty of obtaining, matching footprints when comparing flux measurements from different instruments.

On August 25, the local RFM produced a good match with the tower, in contrast to the (distributed) FFM. Plotting the entire set of RFM fluxes from 25.18:00 yielded a surprise (Fig. 6), where the tower appears to be in a local hot spot. This may also be the case on August 27.19:00 where the flux from the wet sedge around the tower is stronger than the FFM flux from the wet sedge measured by the aircraft. Plots of methane flux against the height of airborne measurement and the strength of turbulence ($\sigma_w$) suggested no simple dependence on these. This flight dramatically shows the hazards inherent in relying on point measurements, which are potentially in nonrepresentative locations, to estimate the area-wide flux. Also note in the middle lower panel of Fig. 4 that the flux of 1 $\mu g \cdot m^{-2} \cdot s^{-1}$ at the tower, though isolated in space, was not isolated in time.

On August 13 everything matched. For flight 13.09:30 the wind was light and the mixing strong ($\sigma_w \sim 0.45\,\mathrm{m} \cdot \mathrm{s}^{-1}$). The warm soil produced a strong methane flux, and the methane flux measured at the tower matches the local RFM flux near the tower as well as the FFM flux from the distributed patches of wet sedge. Importantly, both the summer daytime (13.09:30) and autumn evening (25.18:00) flights showed that when there is reasonable overlap between the tower and aircraft footprints, the flux measurements from the aircraft agree with those from the tower adding another level of validation to the aircraft data.

## 3.2 Regional methane fluxes

During August 2013 FOCAL measured methane flux from a variety of ecotopes across the North Slope. There are six flights used in this analysis; four in the day time and two in the evening (1800 - 1900 local time) which were covered individually in the last section. Keeping that discussion in mind, these data are comparable as a set. Based on the tower data, which exhibit strong and regular diurnal cycles of carbon dioxide and latent heat (not shown), methane has a generally weak diurnal cycle. The sharp feature in the tower trace on August 13 (DOY 225) very likely has a diurnal component, but its shape suggests more than just solar input. This discussion, therefore, will focus on the seasonal change and the methane-emission characteristics of the various surface classes (Figs. 4 and 7).

Land cover type varies over the North Slope, so different flights sampled different types of land cover (see Table 1 and Fig. 2). Wet sedge was the most prevalent and thus was sampled on each flight, except for flight 28.10:00 on the morning of August 28.. Other land cover classes such as bare ground, dwarf shrub, and low-tall willow were also observed but in insufficient quantity to calculate a statistically significant flux. Prevalent near the tower site, which was sampled on August 13, 25, 27, were wet sedge, mesic sedge - dwarf shrub, some lakes, the Sagavanirktok (Sag) River, and fresh water marsh. Soil temperatures in mid-August varied by 1.5 °C with a mean soil temperature of 8 °C at 5 cm depth. By the end of August soil temperatures had dropped to a mean of 3 °C. Wet sedge showed the strongest correlation with soil temperature, with fluxes falling from 2.1 $\mu\mathrm{g} \cdot \mathrm{m}^{-2} \cdot \mathrm{s}^{-1}$ on August 13th to less than 0.5 $\mu\mathrm{g} \cdot \mathrm{m}^{-2} \cdot \mathrm{s}^{-1}$ by the end of August. This relationship held true for emissions from the Sag River with emissions falling from 1 $\mu\mathrm{g} \cdot \mathrm{m}^{-2} \cdot \mathrm{s}^{-1}$ to near 0. Wet sedge, followed by the Sag River, had the largest observed flux of any of the land classes sampled during the first half of August. The other land classes have smaller, more variable fluxes on most flights so that surface class alone does not distinguish them. Most likely the true variability, contributing to the large confidence intervals, is caused by heterogeneity within the surface class in sub-surface soil temperature and water table height. However, within that we can still derive a mean flux based on a large regional sample. Once the soil cools, wet sedge shows reduced, though still positive, flux of methane consistent with the other surface classes measured such as mesic sedge and lakes. The Sag River shows close to zero methane flux. Lakes showed no trend. It should be noted, however, that the number of lakes sampled on August 13 was small and the flux variable as indicated by the large 95% confidence interval. While data from the other land classes sampled on August 13 were sparse, emissions from fresh water marsh and tussock tundra during the latter half of August were similar to those from lakes and the two sedge classes.

Airborne measurements made during August, 2013 are consistent with findings from other studies. Olefeldt et al. (2013) reported sites dominated by sedge and wet soils having methane emissions ranging from 0.46 to 1.6 $\mu\mathrm{g} \cdot \mathrm{m}^{-2} \cdot \mathrm{s}^{-1}$ with a median value of 0.87 $\mu\mathrm{g} \cdot \mathrm{m}^{-2} \cdot \mathrm{s}^{-1}$ across multiple permafrost sites. Other studies at single locations fall into this same range.

For example, Harazono et al. (2006) measured methane fluxes from a wet sedge site in Happy Valley, AK during August of 1995 ranging from 0.38 to 1.5 $\mu g \cdot m^{-2} \cdot s^{-1}$ and Sturtevant and Oechel (2013) measured wet sedge near Barrow with emissions of $0.39 \pm 0.03\,\mu g \cdot m^{-2} \cdot s^{-1}$ with short periods of higher emissions up to 1.1 $\mu g \cdot m^{-2} \cdot s^{-1}$. Emissions from mesic-sedge sites near the Sag River, though south of the areas measured by FOCAL, showed fluxes of 0.35 to 0.58 $\mu g \cdot m^{-2} \cdot s^{-1}$ in the first
half of August falling to 0.12 to 0.23 $\mu g \cdot m^{-2} \cdot s^{-1}$ in the second half of August (Harazono et al., 2006).

    Emissions from lakes tend to be more variable than the land classes. Measured emissions from individual lakes ranged from 0.25 to 6.3 $\mu g \cdot m^{-2} \cdot s^{-1}$ across various thermokarst and other lakes in the North Slope (Walter et al., 2007a; Sepulveda-Jauregui et al., 2015). These fluxes are reported as means over a year, so emission rates during short periods of time may be lower or higher for an individual lake. While FOCAL did not sample the same lakes as in the aforementioned studies, during
the flights near the tower multiple passes over the same lakes allow emissions from individual lakes to be measured. On August 13, five lakes were sampled with sufficient frequency to produce a statistically significant flux. The flux for individual lakes ranged from 0 to 2.6 $\mu g \cdot m^{-2} \cdot s^{-1}$ with a mean for all lakes sampled of 0.18 $\mu g \cdot m^{-2} \cdot s^{-1}$. On August 27 four lakes were measured with emissions ranging from 0.09 to 0.18 $\mu g \cdot m^{-2} \cdot s^{-1}$. The mean methane flux from lakes over the period of the flights shows little flux, except for the lakes sampled on the morning flight of August 28. These are in a different area 250
km west of the tower. Those lakes show an aggregate mean of 0.36 $\mu g \cdot m^{-2} \cdot s^{-1}$, the only flux measured from lakes that was statistically significantly positive (Fig. 7). These data are consistent with the rates measured by the above studies.

## 4   Conclusions

The FOCAL campaign during the summer of 2013 showed how methane fluxes could be successfully measured over large regions using airborne eddy covariance measurements from a small, low-flying aircraft. The data were analyzed in the space/time
domain with both a running flux method using traditional eddy covariance, and the flux fragment method (FFM), a varient using a conditional sampling scheme. Other techniques such as wavelet analysis that rely on the frequency domain to look at the same questions would be worth exploring in the future. A comparison of the theory behind FFM with the theory behind the wavelet method is included in Dobosy et al. (2017).

    Comparison of the airborne measurements to those of a tower showed that the data were quantitatively comparable when
there was good overlap between the tower footprint and aircraft footprint. However, along the flight track local conditions dominated the flux especially in the transition season from summer to fall in late August. Comparing wet sedge at the tower site with wet sedge west of the tower showed a factor of two difference in methane emissions during the latter half of August which underscores the importance of regional measurements as fluxes can have large dependence on spatial heterogeneity even over relatively short distances. During the middle of the summer fluxes from wet sedge were more homogeneous across the
area flown.

    Measurements of methane flux over the North Slope of Alaska in August showed a strong correlation with soil temperature consistent with previous studies. Wet sedge showed the highest persistent methane emissions with mean fluxes about 2 $\mu g \cdot m^{-2} \cdot s^{-1}$ in the first half of August falling to 0.2 $\mu g \cdot m^{-2} \cdot s^{-1}$ in the latter half of August. Emissions from the Sag

River showed a similar trend, while other land surface classes were not sampled enough during the first half of August to provide a statistically significant sample. Individual lakes sampled near the tower showed a large range of emissions varying from near 0 to 2.6 $\mu g \cdot m^{-2} \cdot s^{-1}$ consistent with the range of lake emissions reported in the literature.

Aircraft measurements of surface flux can play an important role in bridging the gap between ground-based measurements and regional measurements based on inversion modeling or downwind-upwind differences. While airborne campaigns are generally more costly than ground based measurements, these costs can be minimized by using small aircraft. For areas that are logistically challenging to access, such as the North Slope, airborne eddy covariance presents the easiest and least expensive way to directly measure surface fluxes regionally with large coverage.

## 5 Data Availability

All data are publicly archived at the NSF ACADIS website (https://arcticdata.io) under citation: David Sayres. 2014. Collaborative Research: Multi-Regional Scale Aircraft Observations of Methane and Carbon Dioxide Isotopic Fluxes in the Arctic. NSF Arctic Data Center. urn:uuid:58bddf69-74fe-4a20-958e-4cd23bb6941f.

*Acknowledgements.* This work was supported by NSF Grant 1203583 and data is archived at the ACADIS website. The authors wish to gratefully acknowledge the efforts and exceptional flying of our pilot, Bernard 'Bernie' Charlemagne, without whom these data could not have been collected.

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

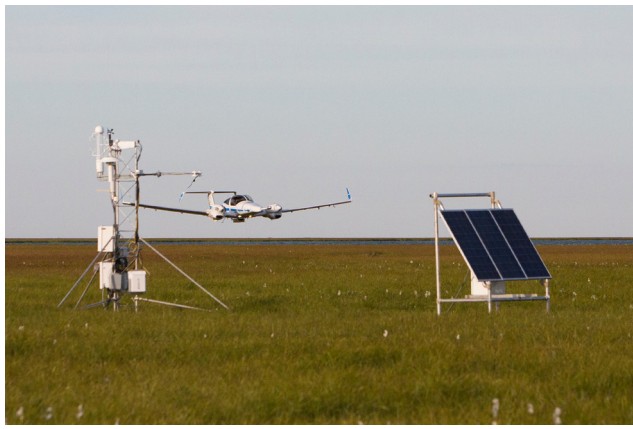

**Figure 1.** Picture of the FOCAL system flying near the NOAA/ATDD flux tower in North Slope, AK.

**Table 1.** Flights used in the analysis along with location, time of day, mean air temperature, and surface land classes.

| Flight Date[1] DD.HH:MM | Location | Start Time UTC-10 | End Time UTC-10 | Temperature[2] (°C ) | Dominant Land Types[3] |
|---|---|---|---|---|---|
| 13.09:30 | Tower | 08:19 | 10:22 | 16 | Sedge, Mesic Sedge, Lakes, Sag River, FWM |
| 25.18:00 | Tower | 17:43 | 19:49 | 5 | Sedge, Mesic Sedge, Lakes, Sag River, FWM |
| 27.11:30 | Western Grid | 09:40 | 13:00 | 6 | Sedge, FWM, Lakes, Tussock tundra |
| 27.19:00 | Tower | 16:46 | 20:02 | 10 | Sedge, Mesic Sedge, Lakes, Sag River, FWM |
| 28.10:00 | Western Grid | 08:39 | 11:39 | 11 | Tussock tundra, Lakes, Lake margins are FWM and Sedge |
| 28.15:00 | Eastern Grid | 13:59 | 15:44 | 16 | Sedge, Mesic Sedge, Lakes, Kuparuk River, FWM |

[1] All flights are during August 2013. DD is the local date of the flight and HH:MM is the middle time of the flight rounded to the nearest half-hour. [2] Temperature calculated as mean temperature recorded by instrument during flight time and below 100 m. [3] Land classes are given in order of largest percent coverage first.

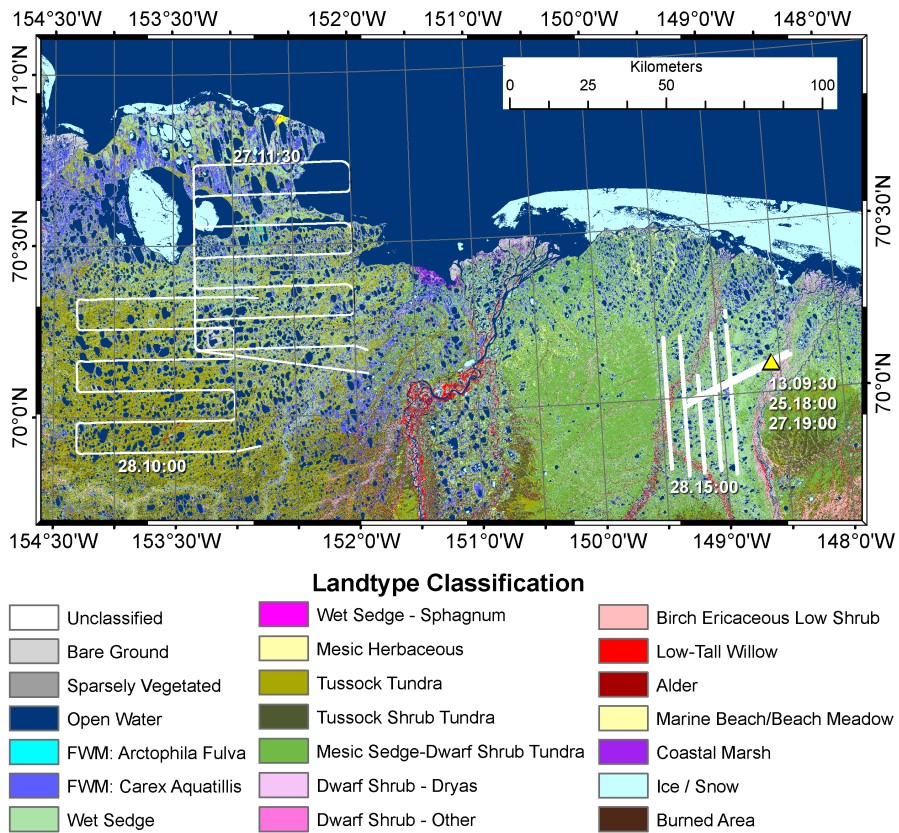

**Figure 2.** Six flight tracks flown by FOCAL during August 2013 are shown in white. Flights are given in the figure as DD.HH:MM, where DD is the date in August and HH:MM is the time (UTC-10hr) of the middle of the flight rounded to nearest half-hour. Flight tracks are shown only for the portions flown within 25 m of the ground. The underlying chart gives the NSSI-assigned land cover produced from LandSat 30-m Thematic Mapper data. The yellow triangle locates the NOAA/ATDD flux tower.

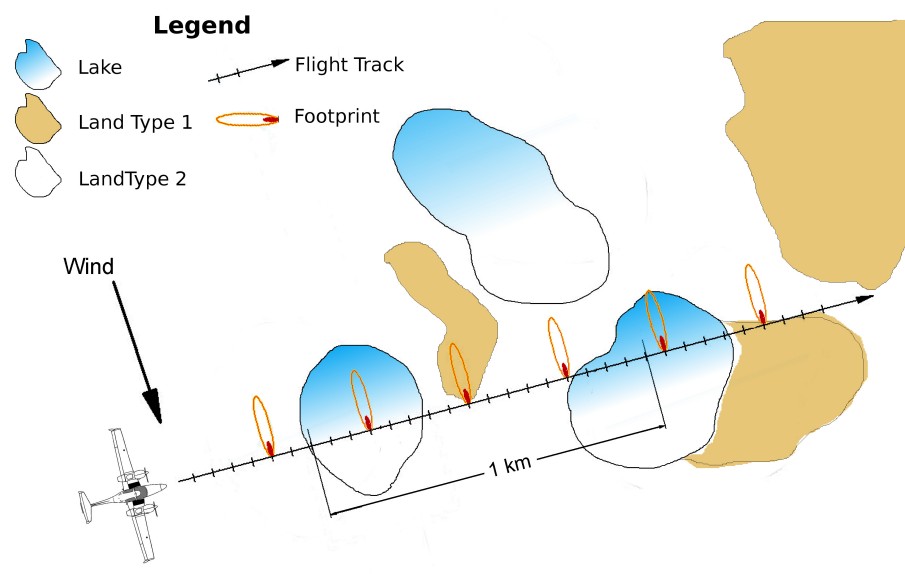

**Figure 3.** The Flux Fragment Method (FFM) divides the covariance measurements into small fragments whose footprints can be attributed to different landscape features or classes. In the figure the landscape has been divided into lakes and two types of land, for example wet sedge and fresh water marsh. Footprints are calculated for each fragment and footprints that lie mostly (>85%) on a single land type are assigned to that land type. All footprints for a given land type can then be summed and divided by the cumulative path length in air.

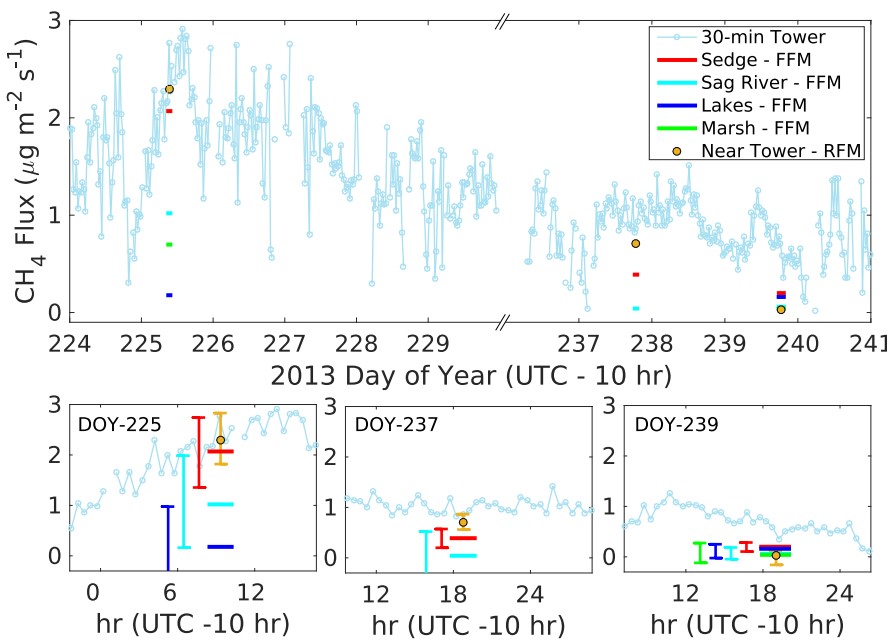

**Figure 4.** Methane flux measured from the flux tower compared with fluxes measured by the FOCAL system. Tower methane fluxes (top plot) are 30-minute means plotted versus day of year. The flights (13.09:30, 25.18:00, 27.19:00 on DOY 225, 237, and 239, respectively) each made repeated passes near the tower. The orange circle gives the mean over these passes of the RFM-determined 3-km flux centered nearest the tower. Fluxes by FFM were aggregated by surface class over the whole flight. The length of the line along the time axis represents the period over which the data were taken, typically 1.5 hr. Lower panels show details for each flight day, labeled by day of year (DOY), with vertical bars showing the 95% confidence interval based on bootstrap analysis. Bars are offset along the x-axis for clarity.

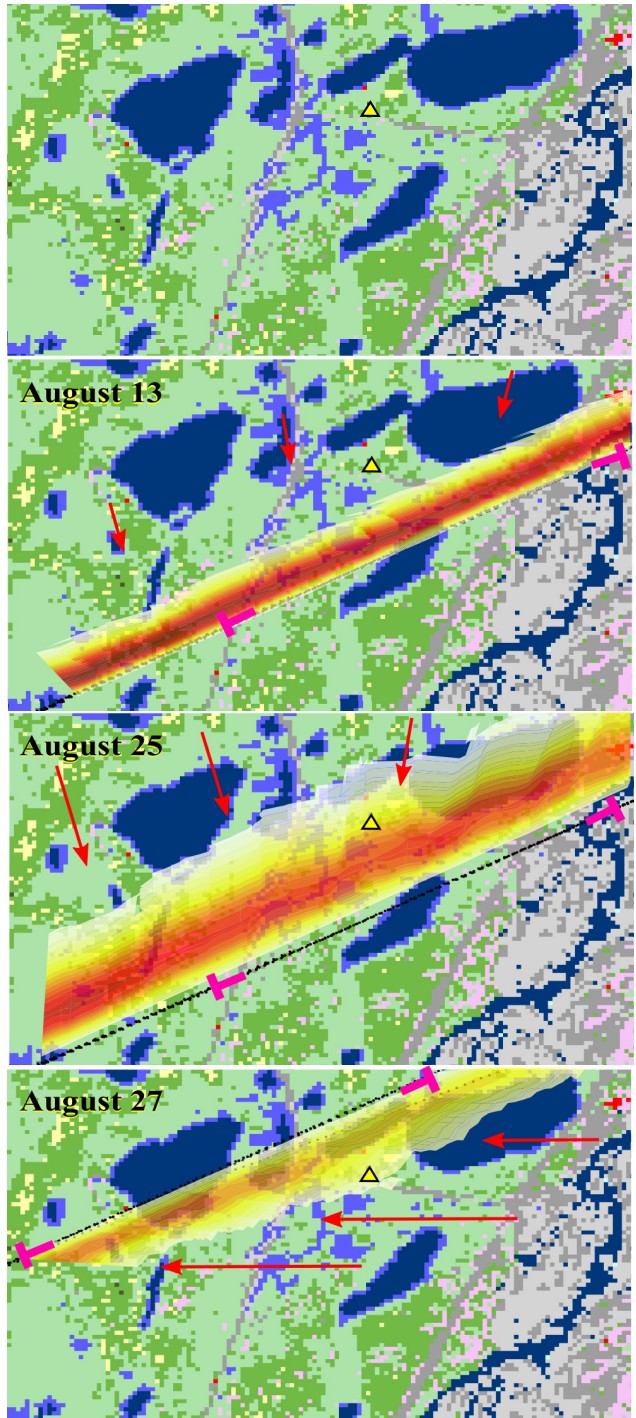

**Figure 5.** Flux footprints near the tower (yellow triangle) for the three tower flights (13.09:30, 25.18:00, and 27.19:00). They are laid over the NSSI-classified land cover map (see Fig. 2). The top panel facilitates identifying the surface classes under each footprint. The flight track, always passing downwind of the tower, is shown as black points, each giving the start position of a flux fragment. The darker and redder ribbon color represents greater probability of contribution to the total flux as described in the text. Red arrows indicate the mean direction of the wind. The part of the flight track used in the near tower RFM calculation is located between the magenta brackets.

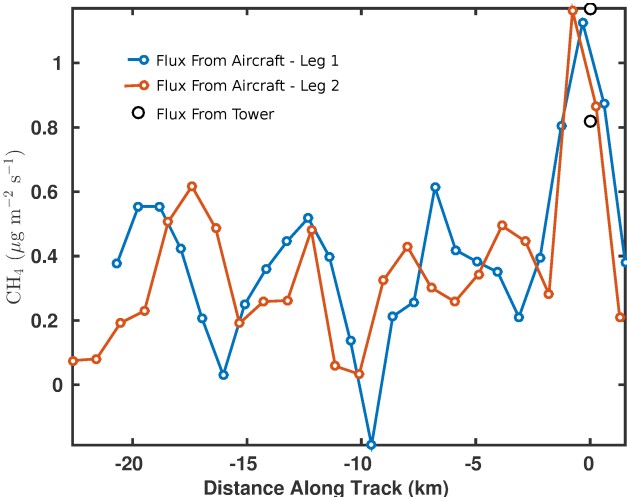

**Figure 6.** Plot of methane flux derived using RFM versus distance from flux tower for two flight legs on August 25. Positive (negative) distance is East (West) of the tower position. The East to West transect (blue) was flown 30 minutes after the West to East transect (orange). Black circles are the methane flux measured by the tower at the nearest time to when the aircraft passed the tower.

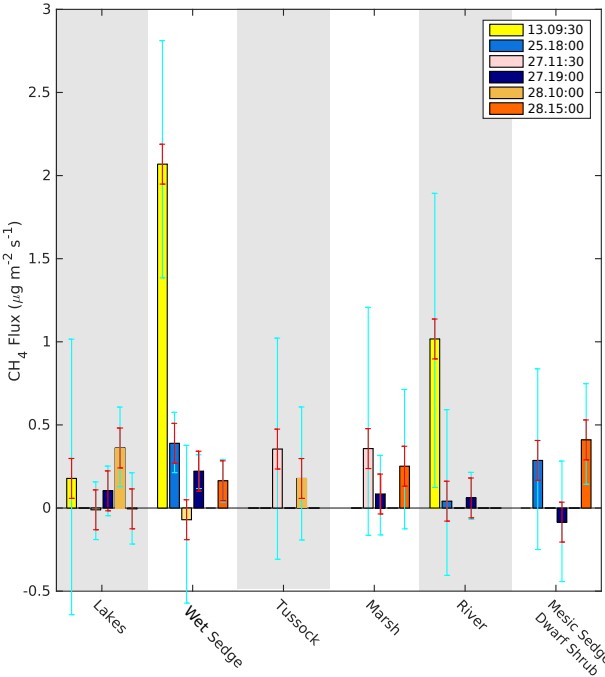

**Figure 7.** Mean methane fluxes by land surface class derived using the FFM for each of six flights as given in the legend. Dates of flights are given as day of month in August followed by the time of the middle of the flight. Bars give the instrument uncertainty (red) and the 95% confidence interval as calculated using bootstrapping (blue).