# Peer review of "Arctic Regional Methane Fluxes by Ecotope as Derived Using Eddy Covariance from a Low-Flying Aircraft"

_Atmospheric Chemistry and Physics, 2016_

## Referee Comment (RC1) · Anonymous Referee #1 · 25 Nov 2016

**Title: "Arctic Regional Methane Fluxes by Ecotope as Derived Using Eddy Covariance from a Low Flying Aircraft"**

**by Sayres et al.**

**General Comments**

The main question addressed by the research is the feasibility of regional-scale airborne measurements of the turbulent methane flux. The goal is to provide land-cover resolved methane fluxes that permit placing continuous tower measurements in a regional context. For this purpose a land cover filter based on space-domain decomposition is applied to the airborne methane flux observations, which are subsequently recomposed by target land cover. The authors conclude that airborne eddy covariance presents an affordable way to directly measure and express surface fluxes across inaccessible regions.

The central questions of the research are interesting and important, and have great potential for closing substantial gaps in the presently available hierarchy of environmental observations. While the presented dataset is a treasure trove, the contribution of the paper to the field is incremental at best. This is due to the application of a conceptually flawed analysis tool to the data:

> The chosen flux fragment method disregards a fundamental micrometeorological underpinning that permits relaxing what is formally a continuity equation to a single "eddy covariance" term: Atmospheric turbulence needs to be sampled from the high-frequency dissipation range to the first low-frequency spectral gap (e.g., Foken, 2008). As mentioned by the authors, the latter would necessitate the analysis of several kilometres of flight data at once. Nevertheless, the authors revert to the use of 60 meters at a time, thus disembodying a few numbers at a time from their theoretical foundation (here: discarding the spatially varying "base state"). Superior space-frequency decomposition techniques are widely available and in use (e.g., Barnhart et al., 2012; Strunin and Hiyama, 2004; Thomas and Foken, 2005; van den Kroonenberg and Bange, 2007). These are not only theoretically sound, but provide better spatial resolution down to meters and do not suffer from the loss of low-frequency contributions.

> Next, relating atmospheric fluxes to discrete land cover classes alone neglects intra-class variability (e.g., Beyrich et al., 2006; Ogunjemiyo et al., 2003). This is better expressed with continuous land cover properties such as temperature, vegetation indices etc. (e.g., Glenn et al., 2008; Ogunjemiyo et al., 1997). In addition, FFM results for individual land covers are not comparable across flight days, as day-to-day synoptic variations and different flight times within the diurnal cycle are not taken into account. Moreover, FFM acts as a filter reducing the use of available data by order 50%, i.e. it is wasteful with respect to data use efficiency. As a result of these shortcoming, Figures 4, 7 show that at a 5% significance level the FFM-derived fluxes do not actually differ between land cover classes, i.e. there is more unexplained variation in the error bars than there is explained variation in the land cover means. Also here, techniques overcoming these systemic deficiencies are available and in use (Jung et al., 2011; Yang et al., 2007)

Hence, to remedy current methodological deficiencies, and to arrive at scientifically more useful and defensible results from this great dataset, I suggest the authors to consider a combination of above methodologies. In fact in their introduction the authors cite Metzger et al. (2013), who demonstrate such combination specifically for the use case of airborne flux measurements.

In consequence, with the current methodology the results only peripherally contribute to the question and do not fundamentally support the conclusions.

The manuscript fits the scope of Atmospheric Chemistry and Physics. The length of the article is appropriate, writing is sufficiently concise and copy-editing is not needed to evaluate scientific merits. While the manuscript is publishable in general I recommend major revisions due to substantial methodological deficiencies.

**Specific Comments**

Please find specific in-text comments in appendix.

**Commented [Reviewer2]:** •Please confirm that you use CH4 dry mole fraction for the covariance / flux calculation.
•In case your calculation is based on partial density, how do you correct for density variations due to temperature and humidity fluctuations (WPL), as well as variations in pressure-altitude and corresponding changes in temperature and pressure, and thus partial density (Poisson equation)?

[Figure]

Aurora Flight Sciences' version of the DA-42, named the Centaur, is a twin-engine aircraft, and has several characteristics that make it an ideal platform for the work discussed here. Due to the twin engine configuration, the entire center fuselage is available for instrumentation and sampling. The Centaur is electrically and structurally well-adapted for carrying a sophisticated scientific payload, having ample spare power from its two alternators and ideally located hard points for the probe and the spectroscopic equipment. Finally, once fixed costs (e.g. aircraft access, instrument integration and certification) have been accounted for, the operating cost of the Centaur are just $1500 per day and $600 per flight hour - a substantial savings compared to many other scientific platforms.

**2.2 Turbulence measurements**

Eddy covariance is a direct way to determine the exchange of mass, momentum, and energy between the atmosphere and the surface. Since the observed flux is assumed to represent the exchange at the surface, the airplane is flown as low as is safely possible, typically below 30 m (Mahrt, 1998). Flux measurements from fixed surface sites, important complements to the airborne measurements, provide extended temporal coverage at selected locations as well as validation of the airborne flux measurements. The covariance of the dry-air mixing ratio of these gases (Webb et al., 1980; Gu et al., 2012) with the turbulent vertical wind component determines the flux as shown in Eq. (1) where the bar represents the average which defines the base state, $\rho_a$ is the dry air density, w' is the departure of the wind from the base state, and c' is the analogous departure of the methane mixing ratio of dry air.

$$F = \overline{(\rho_a w')c'} \tag{1}$$

Unlike a stationary tower, measuring the turbulent vertical wind component from an airplane requires finding the small (vector) sum of the airspeed and the ground speed, two large, nearly canceling vectors. Since both vectors fluctuate rapidly and independently, many independent measurements must be made with precise synchrony at high accuracy and sample rate. Since turbulent fluctuations can be less than 0.1 m/s, the two large velocities must each be accurate within 0.1 m/s. Four samples define the minimum effectively resolvable turbulent eddy size, about 5 m at 60 m/s.

The Centaur uses a small Inertial Navigation System integrated with a GPS (GPS/INS) to report its ground-speed vector over the surface as well as its roll, pitch, and heading, all at 20 Hz. The low-frequency component of the Centaur's velocity is filtered to 1 Hz and extrapolated to the probe's location to mix with the high-frequency component measured directly at the probe. The airspeed vector (airflow relative to the probe) is determined from the distribution of induced pressure over the BAT probe's hemispherical surface. One static pressure and three differential pressures are taken over nine ports. From these four pressure measurements plus temperature come the five relative-flow parameters: ambient pressure, ambient temperature, and three components of the airflow relative to the probe. The dominant airflow component is along the airplane's longitudinal axis, approximately equal to the airplane's true airspeed.

**Commented [Reviewer3]:** •At 5 m above ground this is approximately the eddy wavelength contributing most to the turbulent vertical transport.
•Using the power law of spectral decay, for this platform the need for high-frequency spectral correction of the vertical turbulent flux would be minimal only at measurement heights of 50 m above ground and higher.
•As you are focusing on measurements below 25 m above ground, which high-frequency spectral correction did you use, and how large was the correction?

[revised manuscript text omitted]

**Commented [Reviewer4]:** •Correct, this method neglects low-frequency contributions to the vertical turbulent flux.
•As mentioned above, Ogive analysis typically saturates at 100 - 1000 x measurement height.
•How did you correct low-frequency loss and how large were the contributions to the flux?

**Commented [Reviewer5]:** •How was the turbulence statistics for a robust application of the flux footprint model calculated? A 1 s flux fragment has far too large random error to assume upstream isotropy of the wind field.

**Commented [Reviewer6]:** •This model is 1-D and does not resolve the cross-wind distribution of the influence area – how did you take this into account?
•An updated 2-D version of this model is available (Kljun et al., 2015). Why was this model not used?
•Kljun, N., Calanca, P., Rotach, M. W., and Schmid, H. P.: A simple two-dimensional parameterisation for Flux Footprint Prediction (FFP), Geosci. Model Dev., 8, 3695-3713, doi:10.5194/gmd-8-3695-2015, 2015.

[revised manuscript text omitted]

**Commented [Reviewer8]:** •There are more intuitive ways to visualize the footprint influence area. I am wondering why the authors did not use standard contour plots. Also, it is not apparent from the display whether cross-wind dispersion has been taken into consideration – the individual sequences of dots simply extend in the along-wind direction, which is only half the truth.

[Figure]

[Figure]

Figure 6. Plot of methane flux derived using RFM versus distance from flux tower for two flight legs on August 25. Positive (negative) distance is East (West) of the tower position. The East to West transect (blue) was flown 30 minutes after the West to East transect (orange). Black circles are the methane flux measured by the tower at the nearest time to when the aircraft passed the tower.

**Commented [Reviewer9]:** •Nice display of the tower being biased high in terms of spatial representativeness.

[Figure]

Figure 7. Methane fluxes by land surface class derived using the FFM for each of six flights as given in the legend. Dates of flights are given as day of month in August followed by the flight number for that day. Bars give the instrument uncertainty (red) and the 95% confidence interval as calculated using bootstrapping (blue).

---

## Referee Comment (RC2) · Anonymous Referee #2 · 19 Dec 2016

Sayres et al. present measurements of methane fluxes in the Arctic using a low-flying research aircraft. This dataset is valuable to understand methane emissions in the arctic area and the distribution of methane emissions in different regions. The comparison between aircraft and tower measurements is also encouraging for the usage of airborne flux measurements. The authors used a method called flux fragment method to explore the heterogeneity of the fluxes. But this method is questionable, as each flux calculation only consider data points in a very short period (1 s) and low frequency parts of the fluxes are totally ignored in the calculation. Thus, all of the conclusions made from this part are not justified. As pointed by the other reviewer, other promising methods are available for investigate heterogeneous fluxes, such as wavelet analysis.

A few recent papers have used the wavelet analysis method to determine fluxes of air pollutants in urban and oil/gas regions (Karl et al., 2009;Vaughan et al., 2015;Yuan et al., 2015). The authors are encouraged to try this method. The authors spent some time to introduce the fast measurement system of wind and CH4. Could you add some spectral analysis for measured data.

Figure 4. Can you show the graph as 2*2 layout? The inserts are somewhat misleading and are hard to follow at present layout. Figure 2 and Figure 7: Could you use a consistent way to indicate flight numbers conducted at the same days. Please include this information in the figure caption.

I suggest a major revision is needed before the manuscript can be accepted by ACP. A point-to-point review is still necessary after the revision from the authors.

References:

Karl, T., Apel, E., Hodzic, A., Riemer, D. D., Blake, D. R., and Wiedinmyer, C.: Emissions of volatile organic compounds inferred from airborne flux measurements over a megacity, Atmospheric Chemistry and Physics, 9, 271-285, 2009. Vaughan, A. R., Lee, J., Misztal, P., Metzger, S., Shaw, M. D., Lewis, A. C., Purvis, R., Carslaw, D., Goldstein, A., Hewitt, C. N., Davison, B., Beevers, S. D., and Karl, T.: Spatially resolved flux measurements of NOx from London suggest significantly higher emissions than predicted by inventories, Faraday Discuss., 10.1039/c5fd00170f, 2015. Yuan, B., Kaser, L., Karl, T., Graus, M., Peischl, J., Campos, T. L., Shertz, S., Apel, E. C., Hornbrook, R. S., Hills, A., Gilman, J. B., Lerner, B. M., Warneke, C., Flocke, F. M., Ryerson, T. B., Guenther, A. B., and de Gouw, J. A.: Airborne flux measurements of methane and volatile organic compounds over the Haynesville and Marcellus shale gas production regions, Journal of Geophysical Research: Atmospheres, 120, 6271-6289, 10.1002/2015JD023242, 2015.

---

## Author Comment (AC1) · 27 Jan 2017

The comment was uploaded in the form of a supplement:
http://www.atmos-chem-phys-discuss.net/acp-2016-862/acp-2016-862-AC1-supplement.pdf
* * *

---

## Author Comment (AC2) · 27 Jan 2017

***Final author response to referee comments on*** **"Arctic Regional Methane Fluxes by Ecotope as Derived Using Eddy Covariance from a Low Flying Aircraft" by David S. Sayres et al.**

1. ***Comment from Referee 1:*** *The chosen flux fragment method disregards a fundamental micrometeorological underpinning that permits relaxing what is formally a continuity equation to a single "eddy covariance" term: Atmospheric turbulence needs to be sampled from the high-frequency dissipation range to the first low-frequency spectral gap (e.g., Foken, 2008). As mentioned by the authors, the latter would necessitate the analysis of several kilometres of flight data at once. Nevertheless, the authors revert to the use of 60 meters at a time, thus disembodying a few numbers at a time from their theoretical foundation (here: discarding the spatially varying "base state").*
***Comment from Referee 2:*** *The authors used a method called flux fragment method to explore the heterogeneity of the fluxes. But this method is questionable, as each flux calculation only consider data points in a very short period (1 s) and low frequency parts of the fluxes are totally ignored in the calculation.*

   1.1. **Author response:** A common misconception about the Flux-Fragment Method (FFM) is that the fragments contain no information on scales larger than their length (FOCAL used 60 m). To be sure, fragments formed using departures from local 60-m averages would jettison all larger-scale contributions, but these fragments use departures from the 3-km base state not local averages. Sections 2.2 and 2.3 as written described the method correctly, but insufficiently emphasized this point. The scale of the base state is determined by ogive analysis (Foken, 2008) to be an upper limit for the turbulence present at the time of measurement. The fragments therefore contain information on all scales from the Nyquist wavelength of the sample rate up to the 3-km scale of the spectral gap determined from the ogive analysis. Yet, the air packets quantified by the fragments are also short enough to have likely interacted with a single class of surface. All fluxes defined in the paper are formed from sums of at least 50 fragments, enough to have a cumulative length of at least 3 km, usually more.

   1.2. **Changes to manuscript:** A statement about how the base state for the fragment is made will be added to Page 6, Line 28. "Departure quantities used to form the fragments are relative to a base-state of 3-km scale or more, a scale determined by ogive analysis (Foken, 2008) to be an upper limit for the turbulence present at the time of measurement. The fragments therefore contain information on all scales from the Nyquist wavelength of the sample rate up to the 3-km scale of the spectral gap determined from the ogive analysis. Yet, the air packets quantified by the fragments are also short enough to have likely interacted with a single class of surface."
   Add Page 7, Line 4. "Fluxes are calculated only for those surface class groups whose total length is greater than 3 km."

2. ***Comment from Referee 1:*** *Superior space-frequency decomposition techniques are widely available and in use (e.g., Barnhart et al., 2012; Strunin and Hiyama, 2004; Thomas and Foken, 2005; van den Kroonenberg and Bange, 2007). These are not only theoretically sound, but provide better spatial resolution down to meters and do not suffer from the loss of low-frequency contributions.*
***Comment from Referee 2:*** *As pointed by the other reviewer, other promising methods are available for investigate heterogeneous fluxes, such as wavelet analysis.*

   2.1. **Author response:** The space-frequency decomposition techniques mentioned by reviewer 1 are based in well-developed mathematical theory. Such work as Farge (1993) and Torrence and

Compo (1999) have made multi-resolution continuous-wavelet transforms highly useful to the treatment of turbulence in general and atmospheric turbulence in particular. However, eddy covariance is also theoretically sound and has long been treated successfully (Foken (2008)). The eddy covariance approach remains useful in providing a different, more directly intuitive, physical perspective in the space/time domain. The FFM is a modification of the canonical eddy-covariance. It is unorthodox in its use of conditional sampling to pluck individual fragments from the data stream at will to be combined into a mean covariance. This produces gaps not normally tolerated in space/time eddy-covariance work. The traditional analysis takes advantage of the autocorrelation of the data stream. This advantage is to some extent sacrificed in the FFM, but a large-enough random sample of departure quantities, defined as in response 1.1 above, will produce a meaningful estimate of the flux on all scales of turbulence present in the boundary layer.

Procedures exist to estimate the uncertainty in averages computed over a serially correlated, unevenly spaced data stream (eg. Mudelsee, 2010, Chapter 3).

So long as any significant secondary circulations are accounted in the base-state, the turbulent atmosphere on all its scales can be postulated to repeat over the landscape in a fairly random fashion. A contiguous sample (i.e, without gaps) should not therefore be required. The sample only need be sufficiently large to include multiple instances of boundary-layer structures at each scale. An aircraft moving at airspeed 60 m s$^{-1}$ covers 216 km in an hour encountering 72 instances of 3-km turbulence structure. A sufficiently prevalent class of land surface, whether found in large or small patches is very likely to provide a sufficient sample. Samples which are too short can be discovered in confidence intervals developed by bootstrap resampling as was done by Kirby et al (2008). A more sophisticated bootstrap procedure developed in conjunction with analysis of these 2013 data follows Mudelsee (2010, Chapter 3). A manuscript describing the approach in detail has been submitted to the Journal of Atmospheric and Oceanic Technology and is in review.

In drawing randomly spaced samples from an autocorrelated data series the FFM does sacrifice some efficiency. A contiguous series (or a multi-scale wavelet reconstruction thereof) can take advantage of whatever coherency is contained in the feature it is sampling, though it still must sample several such features to provide an adequate estimator of the mean turbulence and flux found in the study area.

The FFM in the space/time domain is a statistical approach as opposed to decomposition approaches like the wavelet and Empirical-Mode decompositions. Also, being totally in the space/time domain FFM can in principle provide spatial resolution down to whatever scale is required so long as the small-scale features are repeated sufficiently often. Of course, the range of practically realizable scales will depend on the instrumentation used to make the measurements which is the same for wavelet analysis.

Therefore, we see no justification for the referee's conclusion that wavelet analysis is superior to eddy covariance/FFM.  The FFM provides the same spatial resolution, and it does not suffer from the loss of low-frequency contributions suggested by the reviewer.

2.2. **Changes to manuscript:** We do not agree with the reviewers that a change in methodology is

needed. While it might be interesting for a future paper to compare the various approaches and their results, that is not the intent of our present manuscript.

3. ***Comment from Referee 1:*** *Next, relating atmospheric fluxes to discrete land cover classes alone neglects intra-class variability (e.g., Beyrich et al., 2006; Ogunjemiyo et al., 2003). This is better expressed with continuous land cover properties such as temperature, vegetation indices etc. (e.g., Glenn et al.,2008; Ogunjemiyo et al., 1997).*

   3.1. **Author response:** "Better expressed" is a relative assessment, dependent on the question being asked. One may in fact want to determine the intra-class variability to assess the representativeness  of a surface site located in a particular land-use or land-cover class identifiable by remote sensing. Intra-class variability is expressed in our results by the confidence intervals, which as long as the number of fragments is large, mostly represents the variability within that class. One of the goals of the paper is to compare with towers and other published measurements which classify methane flux based on surface classes similar to those used in this paper.

   Specifically, the reviewers' suggestion to use NDVI would be inappropriate for methane measurements. It works somewhat well for $CO_2$ flux because $CO_2$ has a known causal relationship with photosynthesis and plant respiration. Methane is  not primarily controlled by the physiology of the vegetation. Vegetation type may, however, serve as a proxy reflecting different soil moisture and other properties. Also  the roots of sedge are known to act as a passive transport for methane bypassing any oxidation that might otherwise occur in the surface soil. Perhaps a different interval quantity can provide a meaningful correlation to methane flux, e.g. soil moisture, water-table height, or (sub-canopy) soil temperature. These are hard to measure remotely, especially with the accuracy needed. They were not available during the mission, nor do the authors know of a way to do this remotely at the spatial scale necessary. Failing that, we are using surface cover as a proxy for subsurface hydrology. Ignoring any assumptions about subsurface features, our results still show what sort of surface cover is associated with the strongest methane flux. We found wet sedge to dominate $CH_4$ emission when the soil was warm. In particular, it was much more important than open water such as thermokarst lakes, which have garnered much attention based on the work of Walters-Anthony and others.

   3.2. **Changes to manuscript:** modify sentence Page 7, Line 26. "These classifications, assigned based on remotely sensed data, are plausible proxies for properties that have been shown to be primary drivers of methane production and emission such as water table height, soil temperature, and emission pathways such as sedge roots. Interval quantities sensible remotely such as NDVI, air temperature, or other vegetative indexes which correlate with carbon dioxide do not correlate with methane (Olefeldt, 2013). Vegetation classifications such as these have been shown to be useful for estimating regional methane emissions from other regions (eg. Schneider, 2009) though those were based on upscaling from ground measurements."

   Page 6, line 23. "The Flux Fragment Method (FFM) was conceived to answer questions concerning the homogeneity of land classes defined by some remotely sensible measurement in areas where the land classes vary on lengths short compared to what would be needed for a traditional running flux calculation. "

Page 3, line 14. "How representative were towers' footprints of the class of land cover, as identified by remote sensing, in which they were placed? In principle a stationary site can measure all manner of properties and state variables in the soil, the vegetation, and the air within and above the canopy. Much can be learned about the bacteria, soil chemistry, canopy storage, and other quantities relevant to the exchange of mass, momentum, and energy with the surface. But all of this is known only at the one site. How representative is that site of other locations that to remote sensors appear similar? Are there land-cover types that are particularly indicative of emission of a given trace gas? Can the class so identified be used as a quantitative predictor of a particular type of soil chemistry. This is relevant in assessing the regional methane emission from remote sensing. Methane in particular has a fairly complex chemistry in the soil involving state quantities such as the (sub-canopy) soil temperature and the height of the water table. These are measurable only in situ so that having a proxy indicator such as vegetation cover would be valuable.

Aircraft, though more limited in what they can measure than fixed sites, are very mobile providing the opportunity to sample many instances of the same remotely sensed class over the landscape. From this multi-instance sample one can assess how representative the single fixed site is. One can also assess the strength of the variability within the given land-surface class for later investigation from the surface. In remote parts of the earth, in particular, a determination of near homogeneity of emission properties from many instances of a recognizably similar surface class can save considerable effort over a surface-based survey. Alternatively, large variation within a class that is not well predicted by some practically measurable interval quantity will be seen as requiring additional effort for in-situ measurements to find an effective monitoring program for methane emission from that surface class."

Page 7, Line 21 "The questions to be answered by the FFM, using a fuzzy-logic approach (Nguyen and Walker, 2000) to assign surface classes to fragments and then to conditionally sample them based on those classes include:
a) What is the mean flux over all measured instances of each surface class?
b) What surface classes dominate the methane emission, and by how much?
c) How much does the flux over each class vary? Is there a spatial pattern to the variation. The variability will come both from the prevailing atmospheric environment and the heterogeneity of the emission within the same class.
d) How representative is a particular instance of all similar instances over the landscape?"

4. ***Comment from Referee 1:*** *In addition, FFM results for individual land covers are not comparable across flight days, as day-to-day synoptic variations and different flight times within the diurnal cycle are not taken into account.*

4.1. **Author response:** This comment has nothing to do specifically with FFM because the same question could be asked about eddy covariance in general from a tower or aircraft. Again, FFM is a specific implementation of eddy covariance. Synoptic variations are ideally removed by the base state, except as they affect the turbulence. Unlike CO2, methane has a weak to non-existent diurnal cycle. Our tower data do show a weak cycle, most likely caused by near surface soil temperature changes through the day. However, this diurnal cycle is an order of magnitude less than the variation due to other causes including deeper-soil temperature. It is also smaller than the difference observed between surface classes and therefore comparing flights even though

they were at different times of day is justified.

4.2. **Changes to manuscript:** Page 10, Line 4 add "Though some of the flights were in the evening (1800 – 1900 local time) and some in the morning, these data are still comparable. Unlike CO2, methane has a weak to non-existent diurnal cycle (Figure 4). Based on our tower data we do show a very weak cycle, most likely caused by near-surface soil-temperature changes through the day. However, this diurnal cycle is much weaker (<0.2 ug m-2 s-1) than the class to class variations, seasonal variations, or variations due to other factors. Therefore, comparing flights even though they were at different times of day is justified. The sharp feature in the tower trace on August 13 (DOY 225) probably has a diurnal component, The important comparison, however, is between the strong methane flux in the summer regime of first half of August and the much weaker flux in the autumn regime of later August after the major reduction in soil temperature."

Figure 7 has been altered to give better evidence of which flights occurred in the daytime and which in the evening.

[Figure]

5. ***Comment from Referee 1:*** *Moreover, FFM acts as a filter reducing the use of available data by order 50%, i.e. it is wasteful with respect to data use efficiency.*

5.1. **Author response:** The FFM retains all data suitable for flux calculation. The data are simply

stored and used as 60-s sums of cross products, a convenient form flexible enough to allow many different treatments. The particular approach used in this paper selects a subset of these fragments to address the question being asked, which is to identify discrete land-cover classes that stand out in their contribution to landscape-wide emission of methane. The focus of the current analysis is to examine the spatially dominant land classes in their "pure" form, so rather stringent criteria were applied which, it is true, removed about half of the fragments from the analysis. The FFM was conceived to answer this question: how representative is a single fixed site of other locations on a heterogeneous surface that to remote sensors appear similar? How good is the land-cover class occupied by that site as a proxy for methane flux? The more representative of a single land class the fragment is, the more significant the differences between land classes becomes.

The FFM, however, is not limited to addressing this question alone. Fragments could just as well be associated with values of some interval quantity such as a carbon-isotope ratio, NDVI, or the fraction of footprint occupied by each of several land classes. For this study we wanted to compare to other published measurements and assess the intra-class variability. Limiting the results to a few well sampled classes was better suited to that purpose.

5.2. **Changes to manuscript:** No change.

6. ***Comment from Referee 1:*** *Figures 4, 7 show that at a 5% significance level the FFM-derived fluxes do not actually differ between land cover classes, i.e. there is more unexplained variation in the error bars than there is explained variation in the land cover means. Also here, techniques overcoming these systemic deficiencies are available and in use (Jung et al., 2011; Yang et al., 2007)*

6.1. **Author response:** 1. Not all land-surface types emit significantly different amount of methane. This is not a problem given the questions we are asking. And while many land classes have similar methane emissions, others have significant difference, e.g. mesic sedge and wet sedge on August 13, or lakes and wet sedge on August 13. For the most part, after the soil cooling, the various land classes are not distinguished in their methane release. Keep in mind the goal is not to come up with a criterion that distinguishes land class by its methane emission (or to predict land class based on methane emissions), but to measure regionally aggregated methane emissions from each of a limited number of land classes. It is reasonable that some land classes will have similar methane emissions, especially for land classes that emit little methane.

2. Broader confidence intervals reveal lower statistical power. Typically for our data set, the broader confidence intervals are associated with the shorter samples (which reduces the power). A statistical sample, to the extent that it is independent and identically distributed is a repeated drawing from the population. If a particular outcome happens only 5% of the time, then at each drawing it has a 5% chance of being realized. But with repeated drawing, the chance increases of getting at least once some outcome having a 5% chance or less. In a very large sample, each outcome having a 5% chance will occur 5% of the time. But more than 5% of a small sample will comprise some outcomes individually having a 5% chance. If one uses a bootstrap method, which assumes the realized sample to be the entire population, a disproportionate number of population members will be outcomes that in the full population would be much less likely to occur. Of course, a new measurement set will contain a comparable number of unlikely outcomes, but they will be different from those in the earlier set of measurements. Adding new

data thus reduces the overall likelihood of all low-probability events and increases the power. Unfortunately, getting a new set of measurements is **expensive.** So the tails of the distribution developed using a relatively short sample of actual measurements will be biased toward greater probability than the true population. It will therefore have wider confidence intervals (which depend on the weakness of the tails) than would the true population. Techniques have been developed to address this issue, but their implementation is not trivial. They belong to the next generation of the FFM.

3. Our measurement of wet sedge has the greatest power, Second greatest is often lakes, but may be another land-cover type. Sedge is a strong emitter, but its confidence interval is shorter in part because we have a longer sample from it.

6.2. **Changes to manuscript:** Page 10, Line 23 add "Wet sedge, followed by the Sag river, had the largest observed flux of any of the land classes sampled during the first half of August. The other land classes have smaller, more variable fluxes on most flights so that surface class alone does not distinguish them. Most likely the true variability, contributing to the large confidence intervals, is caused by heterogeneity within the surface class in sub-surface soil temperature and water table height. However, within that we can still derive a mean flux based on a large regional sample. Once the soil cools, wet sedge shows reduced, though still positive, flux of methane consistent with the other surface classes measured such as mesic sedge and lakes. The Sag river shows close to zero methane flux.

Page 11, Line 9 add "The mean methane flux from lakes sampled on a flight by flight basis shows little flux on average, except for the lakes sampled on 130828.3, which are in a different area 250 km west of the tower. Those lakes show an aggregate mean of 0.36 ug m-2 s-1 (Figure 7)"

7. *Comment from Referee 1: I suggest the authors to consider a combination of above methodologies. In fact in their introduction the authors cite Metzger et al. (2013), who demonstrate such combination specifically for the use case of airborne flux measurements.*
*Comment from Referee 2: A few recent papers have used the wavelet analysis method to determine fluxes of air pollutants in urban and oil/gas regions (Karl et al., 2009;Vaughan et al., 2015;Yuan et al., 2015). The authors are encouraged to try this method.*

7.1. **Author response:** These papers look promising as discussions of how one can operate in urban and fracking regions. As explained in the first few responses, the FFM is a reasonable and sound method for analyzing these data. Its value derives from its position as an alternate approach from a different perspective (space/time domain). A comparison of the different methodologies is an activity we hope to pursue, but that is outside the scope of this present paper.

7.2. **Changes to manuscript:** No change.

8. *Comment from Referee 1: This hemispherical model requires calibration, in the case of very low-level flight in particular to offset dynamic upwash and ground effect which otherwise affect the covariance calculation (e.g., Crawford et al., 1996; Garman et al., 2008). Have these calibrations and corrections been performed, and if so to within which residual error?*

8.1. **Author response:** We flew multiple calibration maneuvers both in preparing for and during the Alaska campaign. Before assembling the FOCAL system, we characterized the BAT (gust) probe in a wind tunnel (Dobosy et al., 2013). We also tested a similar BAT probe in flight on a different aircraft (Vellinga et al., 2013, hereafter V2013). After calibration derived from a flight taken on the evening of August 27 in Alaska, we performed the yaw maneuver described by V2013 and obtained a residual contamination within 10%, as described there. A pitch maneuver described by V2013 was performed resulting in contamination of 10% for the high-frequency pitching (1.6 s period), which was the best executed of the pitch test's three parts and is the severest test.

8.2. **Changes to manuscript:** Page 4, Line 9 insert new paragraph "Before assembling the FOCAL system, we characterized the BAT probe in a wind tunnel (Dobosy et al., 2013). We also tested a similar BAT probe in flight on a different aircraft (Vellinga et al., 2013, hereafter V2013). After the FOCAL system was assembled, similar calibration maneuvers were flown in preparation for and during the Alaska campaign. As part of a calibration flight on the evening of August 27 in Alaska, we performed the yaw maneuver described by V2013 and obtained a residual contamination within 10%, as described there. A pitch maneuver described by V2013 was performed resulting in contamination of 10% for the high-frequency pitching (1.6 s period), which was the best executed of the pitch test's three parts and is the severest test.

9. ***Comment from Referee 1:*** *Please confirm that you use CH4 dry mole fraction for the covariance / flux calculation. •In case your calculation is based on partial density, how do you correct for density variations due to temperature and humidity fluctuations (WPL), as well as variations in pressure-altitude and corresponding changes in temperature and pressure, and thus partial density (Poisson equation)?*

9.1. **Author response:** We had provided this confirmation in the manuscript, page 4, line 25, and also on page 5, line 13 citing both Webb et al. (1980) and its update, Gu et al., (2012). We will move this citation back to the first mention of the gas measurements.

9.2. **Changes to manuscript:** Page 4, Line 25, add (Webb et al., 1980; Gu et al., 2012).

10. ***Comment from Referee 1:*** *At 5 m above ground this is approximately the eddy wavelength contributing most to the turbulent vertical transport. •Using the power law of spectral decay, for this platform the need for high-frequency spectral correction of the vertical turbulent flux would be minimal only at measurement heights of 50 m above ground and higher. •As you are focusing on measurements below 25 m above ground, which high-frequency spectral correction did you use, and how large was the correction?*

10.1. **Author response:** Plots of the spectra and cospectra of the data streams of vertical air motion and the dry-air mixing ratios of the trace gases were prepared and presented in a paper that was submitted to J. Ocean. Atmos. Tech. We have not used high-frequency spectral corrections as long as the highest wavenumber for vertical wind was clearly in the inertial subrange, i.e. following the -5/3 power of the wavenumber, and clearly above the wavenumber of the maximum spectral density. A data-starvation test using the flux runs from the evening of August 25 yielded an estimated loss of about 10% in fluxes computed with a coarser sample rate. Presenting a long discussion of the the spectra and cospectra seemed out of scope for the current paper. A discussion is included in a separate paper submitted to the Journal of Atmospheric and

Oceanic Technology (JTECH).

A regression of 3-km running flux (see Section 2.3.1) against the height above ground for flight 13.09:30 was run to assess the correlation of flux with altitude. A quadratic regression was required yielding significant positive slope but significant negative curvature. The regression line reached a maximum at an intermediate point before the maximum height above ground. Furthermore, the regression explained only 10% of the variance.

10.2. **Changes to manuscript:** add above to Page 4, line 9 and after additions from 8.2 above.

11. ***Comment from Referee 1:*** *Correct, this method neglects low-frequency contributions to the vertical turbulent flux. •As mentioned above, Ogive analysis typically saturates at 100 - 1000 x measurement height. •How did you correct low-frequency loss and how large were the contributions to the flux*

11.1. **Author response:** This comment was addressed in the response the reviewers' objection 1 above.

11.2. **Changes to manuscript:** See changes from comment one above.

12. ***Comment from Referee 1:*** *How was the turbulence statistics for a robust application of the flux footprint model calculated? A 1 s flux fragment has far too large random error to assume upstream isotropy of the wind field.*

12.1. **Author response:** The turbulent statistics required to parameterize the model of Kljun et al. (2004) were computed from averages taken over the length of each flight leg, where the flight leg was defined as the straight segment between turns over which the collected data were used. The detrending (subtracting the base state from the original series) was done over each flight leg. Typically the flight legs were 15 km to 20 km.

12.2. **Changes to manuscript:** Page 7, Line 17  [Start in 13.2] "We use the parameterization scheme described in Kljun et al. (2004) which uses a backward Lagrangian model (Kljun et al., 2002) for a range of heights, stability measures and other turbulence quantities that are measured from the aircraft. The turbulence quantities  are computed from averages taken over the length of each flight leg, where the flight leg is defined as the straight segment, between turns, over which the collected data were used. [continue at 13.2, second part]

13. ***Comment from Referee 1:*** *This model is 1-D and does not resolve the cross-wind distribution of the influence area – how did you take this into account? •An updated 2-D version of this model is available (Kljun et al., 2015). Why was this model not used?*

13.1. **Author response:**
Since we use the surface class as a categorical quantity the crosswind-integrated form of the footprint model of Kljun et al. (2004, KCRS04) was considered appropriate for our use as a membership function for the fuzzy set (Nguyen and Walker, 2000) of a particular surface class. The selected 85% membership criterion is strict so as to admit only particularly representative instances of the surfaces encountered.

The more recent work of Kljun & Co. (2015, KCRS15) became known to us late in our investigation. In providing an explicit crosswind distribution to the footprint it represents significant advance over KCRS04. However, the crosswind-integrated footprint of KCRS04, fundamentally unchanged, provides the backbone for the two-dimensional footprint of KCRS15. The crosswind spread may be important, for example, where an interval quantity, such as NDVI is to be calculated from the footprint of each unit of flux in order to train a regression or machine-learning model, such as done by Metzger & Co, (2013), or Ogunjemiyo & Co (2003).

The present study was not intended to produce a regression scheme. It is about the role of each surface class (as a category) in the emission of methane. Since the footprint is computed every 60 m the procedure will identify all instances of the surface classes present except for the very smallest. Expanding the footprints to two dimensions does not appear to add sufficient value to justify recalculation. The results would be unlikely to produce any changes in the results.

13.2. **Changes to manuscript:** Page 7, Line 14 "Finally, a footprint model is applied to estimate the level of influence of each surface type on each fragment. This provides a measure of membership of that fragment in the fuzzy set (Nguyen and Walker, 2000) associated with each surface type, treated as a categorical variable. Fragments having a sufficient level of membership for a particular surface class are assigned to that class. A membership level above 0.5 restricts all fragments to no more than one class. Fragments can thus be grouped into sets all members of which have a measure greater than a prespecified level of the probability that they came from the same surface type (see sec. 3.2 for examples of how FFM is used to interpret these data)" [continue at 12.2]

[second part, continued from 12.2] The more recent two-dimensional version (Kljun et al., 2015) was not considered necessary because of the footprint's restricted use as a membership criterion to assign a selected subset of fragments to the surface categories.

14. ***Comment from Referee 1:*** *There is no such website. Where can the data (incl. raw data) be accessed?*

14.1. **Author response:** The URL was missing an 's'. Should have been https://. Thanks for pointing this out.

14.2. **Changes to manuscript:** Page 12, Line 2 "https://arcticdata.io"

15. ***Comment from Referee 1:*** *There are more intuitive ways to visualize the footprint influence area. I am wondering why the authors did not use standard contour plots. Also, it is not apparent from the display whether cross-wind dispersion has been taken into consideration – the individual sequences of dots simply extend in the along-wind direction, which is only half the truth.*

15.1. **Author response:** With crosswind integrated footprints, it makes sense to plot them as lines, rather than as 2-D contour plots. However, to show the full footprint area along the flight track we have modified figure 4 to show a ribbon of footprint probabilities for one leg of each flight track for each day. Arrows have been added to show the dominate wind direction, which was observable before from the individual footprints. Hopefully this will be clearer for the reader.

15.2. **Changes to manuscript:** Figure 5 has been modified as described above.

[Figure]

Figure 5. Map of area surrounding the flux tower (yellow triangle) with false color map representing different land classes defined as in Fig. 2. Bottom three plots show three days when data was taken near the tower. The flight track for each flight is shown as black points, where each point is the start position of a flux fragment. Colored ribbon shows the flux footprints along the flight track. The darker and redder color of the ribbon represents larger probability of contribution to the total flux as described in the text. Red arrows indicate the mean direction of the wind.

16. ***Comment from Referee 2:*** *The authors spent some time to introduce the fast measurement system of wind and CH4. Could you add some spectral analysis for measured data.*

16.1. **Author response:** See 10.1

16.2. **Changes to manuscript:** See 10.1

17. ***Comment from Referee 2:*** *Figure 4. Can you show the graph as 2\*2 layout? The inserts are somewhat misleading and are hard to follow at present layout.*

17.1. **Author response:** We have modified Figure 4 by breaking it into four panels. One long panel displays the tower data, locating the three near-tower flights as before. Temporal resolution was improved by displaying only the periods when the aircraft was operating. The three insets have been relocated as individual panels underneath the tower data and are labeled by flight day instead of a,b,c. The abscissa of each is now given as (local) time of day to show the actual time of flight.

17.2. **Changes to manuscript:**

[Figure]

Figure 4. Comparison of methane flux measured by the flux tower with fluxes measured by the FOCAL system. Tower methane fluxes (top plot) are 30-minute means plotted versus day of year. Three flights (Aug. 13, 25, and 27) made repeated flight transects near the tower. A running mean flux, using the nearest 3 km of flight track to the tower for each leg, was calculated and the mean of these fluxes is plotted for each day as an orange circle. Fluxes for wet sedge, marsh, lakes, and the Sag river were calculated using FFM using data from the whole

flight and are plotted for each day, color coded according to the legend, with the length of the line along the time axis representing the time over which the data were taken. Bottom plots show details for each flight day, labeled by day of year (DOY), with bars showing the 95% confidence interval based on bootstrap analysis. Bars are offset along the x-axis for clarity.

18. ***Comment from Referee 2:*** *Figure 2 and Figure 7: Could you use a consistent way to indicate flight numbers conducted at the same days. Please include this information in the figure caption.*

    18.1. **Author response:** It is consistent, Figure 7 just leaves off the common 1308 part, but we can add that back into the figure legend. The information is already included in the captions of fig 2 and 7 and table 1.

    18.2. **Changes to manuscript:** We have modified the date convention to include the flight time and changed, Table 1 and Figures 2 and 7. The new convention uses DD.HH:MM.

References added:

Dobosy, R., E.J. Dumas, D.L. Senn, B. Baker, D.S. Sayres, M.F. Witinski, C.E. Healy, J. Munster. and J.G. Anderson, 2013: Calibration and quality assurance of an airborne turbulence probe in an aeronautical wind tunnel. Journal of Atmospheric and Oceanic Technology, 30 (2), 182–196.

Gioli, B., Miglietta, F., De Martino, B., Hutjes, R. W. A., Dolman, H. A. J., Lindroth, A., Schumacher, M., Sanz, M. J., Manca, G., Peressotti, A., and Dumas, E. J., 2004: Comparison between tower and aircraft-based eddy covariance fluxes in five European regions, *Agricultural and Forest Meteorology*, **127**, 1–16.

Kljun, N., P. Calanca, M.W. Rotach, H.P Schmid, 2015: A simple two-dimensional parameterisation for Flux Footprint Prediction (FFP), *Geoscientific Model Development*, **8(11)**, 3695-3713

LeMone, M., R. Grossman, F. Chen, K. Ikeda, and D. Yates, 2003: Choosing the averaging interval for comparison of observed and modeled fluxes along aircraft transects over a heterogeneous surface. *Journal of Hydrometeorology*, **4**, 179–195.

Metzger, S., Junkermann, W., Mauder, M., Butterbach-Bahl, K., Trancón y Widemann, B., Neidl, F., Schäfer, K., Wieneke, S., Zheng, X. H., Schmid, H. P., and Foken, T. 2013: Spatially explicit regionalization of airborne flux measurements using environmental response functions, *Biogeosciences*, **10**, 2193-2217, doi:10.5194/bg-10-2193-2013.

Mudelsee, M., 2002: TAUEST: A computer program for estimating persistence in unevenly spaced weather/climate time series. *Computers& Geosciences*, **28** (1), 69–72.

Mudelsee, M., 2010: *Climate time series analysis*. Springer, 474 pp.

Nguyen, H.T., and E. A. Walker, 2000: *A First Course In Fuzzy Logic,* Chapman & Hall/CRC, ISBN 0-8493-1659-6, 373 pg.

Ogunjemiyo, S. O., Kaharabata, S. K., Schuepp, P. H., MacPherson, I. J., Desjardins, R. L., and Roberts, D.A. 2003: Methods of estimating CO2, latent heat and sensible heat fluxes from estimates of land cover fractions in the flux footprint, *Agric. For. Meteorol.*, **117**, 125-144, doi:10.1016/S0168-1923(03)00061-3.

Schneider, J., G. Grosse, D. Wagner Land cover classification of tundra environments in the Arctic Lena Delta based on Landsat 7 ETM+ data and its application for upscaling of methane emissions, *Remote Sensing of Environment*, **113**, 380-391, 2009.

Torrence, C., and G. P. Compo, 1998: A practical guide to wavelet analysis. *Bulletin of the American Meteorological Society*, **79** (1), 61–78.

Vellinga, O. S., R. J. Dobosy, E. J. Dumas, B. Gioli, J. A. Elgers, and R. W. A. Hutjes, 2013: Calibration and quality assurance of flux observations from a small research aircraft. Journal of Atmospheric and Oceanic Technology, 30 (2), 161–181.

---

## Author Response (AR2)

***1. Comment from Editor:*** *Your reply 2.1 discusses the FFM to extensively. As the FFM is not well known in the flux community, could you add shortly the key points of this, and about assumptions, into the manuscript,*
*e.g.: -Its statistical approach to flux calculation (somewhat similar to disjunct eddy covariance approach)*
*-Difference to RFM and wavelet approaches*
*Also, it would be good to explicitly discuss the assumptions behind FFM, e.g.:*
*-that there are no systematic flow patterns over different land cover classes)*
*-that the land cover can be divided into distinct, internally homogeneous, land cover types (this relates to point 3 as well*

> **1.1 Author Response:** We have added several sentences to cover the points you mention above. We will emphasize its relation to eddy covariance. Making an analogy to another technique such as disjunct eddy covariance (DEC) is a good idea, but in this case DEC differs from FFM in a number of ways. In either system, the gust probe's measurements are reported at a relatively high rate because of their short sample-acquisition and measurement-response times. DEC applies to chemical sensors and others that can quickly acquire a sample but require a longer time interval to analyze it. That is, the rate-limiting step is the sensor's process time, not the acquisition time of the sample.
> The FFM assumes all instruments acquire and analyze samples at a rate sufficient to permit ordinary eddy-covariance analysis. Then it breaks the covariance sums (not averages) into small fragments that can later be subsampled at will to compute a mean flux density over an ensemble of fragments not necessarily contiguous in time.
> **1.2 Changes to Manuscript:** relation to eddy covariance added page 7, lines 3 and 25. Assumptions about turbulent scales added page 8, lines 6-14.

***2. Comment from Editor:*** *You could also mention that comparing FFM and wavelet is an interesting question but not tackled in this paper.*

> **2.1 Author Response:** A comparison of the theory behind FFM with the theory behind the wavelet method is included in a second paper in review at the JTECH. Analysis of the FOCAL data using the wavelet technique awaits resource availability. We will cite this paper in the manuscript.
>
> **2.2 Changes to Manuscript:** Sentences about comparison to wavelets were added on page 13, line 33 to page 14, line 2.

***3. Comment from Editor:*** *Changes to MS, 6.2: can't find the latter addition from the revised manuscript.*

> **3.1 Author Response:** Lines numbers got changed when the previous section got rewritten and I forgot to update the changes with the new lines numbers. These changes were on page 13, line 8.

***4. Comment from Editor:*** *Reply 10.1: I do not see the reasoning not to do high-frequency correction when the highest measured wavenumber is in inertial sub-range. This is no guarantee that the high frequency losses wouldn't be significant. What you mean by data starvation test, can you describe this in more detail.*

> **4.1 Author Response:** The "data starvation" test measured the loss of reported flux due to a reduced sample rate of the gas sensor down to between 3 $s^{-1}$ and 4 $s^{-1}$. The samples were acquired at the same 10 $s^{-1}$ rate as specified by the design, but where the 10 $s^{-1}$ data stream was

subsampled at an even rate of 3 s$^{-1}$ and at an uneven rate that matched a subset of the data stream from 13.09:30. Flux computed from the stream at the full sample rate was about 10% stronger than the subsampled (data-starved) streams. We did not assess the effect of loss of flux at sample rates higher than 10 s$^{-1}$. Some time ago we tried estimating the additional flux to be expected through the inertial subrange at shorter scales than we had sampled and were not impressed with the results. It appears worth revisiting, but we did not try this for the present study. It appears unlikely that such high-frequency information would contribute to a change in the magnitude of the flux significantly beyond the 10%. One possible exception is 27.19:00, when the spectrum of $w$ barely reached the inertial subrange before its Nyquist frequency.

**4.2 Changes to Manuscript:** Page 5, line 10-14 "To assess the potential loss of high-frequency flux, a data starvation test was done taking two subsamples from a full 10 s $-1$ stream of gas-sensor data. One was evenly spaced at 3 s $-1$ , the other more randomly spaced between 3 s $-1$ and 4 s $-1$ . The flux from the full-rate sample was about 10% stronger than that from either subsampled stream.  This was considered acceptable for the present study. Methods to extend the high-frequency range will be explored for future work."

*5. Comment from Editor: Comment 13, and answers: It would be good to have a comparison on how wide the 2D footprint is compared to typical land cover class fraction. This would maybe justify not using 2D footprint.*

**5.1 Author Response:** We computed a two-dimensional footprint estimate from conditions characteristic of the most convectively active flight of the set (13.09:30) and added the following text to the manuscript

**5.2 Changes to Manuscript:** add lines 7-15 on page 9. "The more recent two-dimensional version (Kljun et al., 2015), which was considered too computationally intensive to be included in the present study, was not considered necessary because of the footprint's current restricted use as a membership criterion to assign a selected subset of fragments to the surface categories. The degree of overlap was assessed, however, for future reference. Using the measurements from the convective daytime case 13.09:30 the two-dimensional model yielded a footprint with a full width of about 250 m ($\pm 1 - \sigma$ y ) at the location of maximum crosswind-integrated probability, 93 m upwind of the sensors. Since the probabilities are weighted towards the middle of the footprint and the land classes tend to be homogeneous on the order of at least 300~m, using the one-dimensional version of the model is acceptable given our focus on categorical classification and our strict membership criterion (85%). With interval quantities the weighted distribution of sources over the full two-dimensional footprints will be required.

Recalculation every 60 m thus implies considerable overlap. With our focus on categorical classification and our strict membership criterion (85%) this overlap is acceptable. With interval quantities the weighted distribution of sources over the full two-dimensional footprints will be required."

*6. Comment from Editor: Furthermore, when calculating the membership to certain class, did you weigh the contribution by footprint function, or did you take certain %-area as footprint area and use that (non-weighed) in membership calculation. I'd assume the former, but couldn't find it explained in the text.*

**6.1 Author Response:** It is the former as you suggest. We moved the discussion of this from section 3.2 and put it into section 2.4 where we first introduce the land classes. It has also been

modified to make clearer how land class membership was assigned.

**6.2 Changes to Manuscript:** A new third paragraph of section 2.4 now reads: "In order to distinguish the contribution to the total methane flux from individual land types and to assess the variability across ecotopes, the data are filtered to only include flux fragments having a membership score of at least 85%, determined by integrating the length of the footprint's centerline weighted by the crosswind-integrated probability density that the flux came from a single surface class. Increasing this threshold increases the link between the calculated flux and a single land class, but reduces the number of footprints available for the analysis thus widening the confidence interval. Varying the threshold between 80% and 95% produces only a small effect on the quantification of flux from each land class. We find that 85% is a good compromise between singling out individual land classes while still retaining a sufficient dataset. For the flight speed of the Centaur at low altitude and wind conditions during the flights, the length of the footprint contributing more than 90% of the flux for each 60-m fragment, varied between 100 and 800 m. The above filter eliminates about a third to half of the flux fragments from each flight. Of those, we limit the land classes to those where the total number of flux fragments is more than 50 fragments or an equivalent distance of 3 km. The flux fragments are summed and then divided by the total integration length for each land cover type (Fig. 7)."

**7. Comment from Editor:** *Page 11, lines 14-19: Here you discuss on the representativity of the tower fluxes based on flights on one day. How about the other days with flight near tower, do they show the same pattern?*

**7.1 Author Response:** The flight on the 13th matches very well between the tower and the sedge measured along the track. The 27th ,as discussed in section 3.1, is harder to tell as the turbulence was weak and the footprints near the tower were really over lakes but seems to show the same pattern as the 25th that flux from sedge in general is half that of the tower. We will add a few sentences to this effect.

**7.2 Changes to Manuscript:** added page 12, line 7 "This may also be the case on August 27.19:00 where the flux from the wet sedge around the tower is stronger than the FFM flux from the wet sedge measured by the aircraft." The 13th is already discussed on page 12, lines 13-17.

**8. Comment from Editor:** *Page 12, lines 4-7: "For the flight speed of the Centaur 5 at low altitude and wind conditions during the flights, the footprint length for each 60-m fragment varied between 0.5 and 5 km, though the part of the footprint whose probability of contributing more than 90% of the flux was between 100 and 800 m long". How do you define the footprint length of 0.5 - 5 km? Theoretically, there is no absolute boundaries of footprint function.*

**8.1 Author Response:** We will just list the length that contributes to 90% of the flux.

**8.2 Change to Manuscript:** Modified sentence on page 10, line 25.

**9. Comment from Editor:** *Page 8, line 17: "We use this to establish transects at altitudes typically 10 m to 30 m above ground; low as safely possible" would be better as "We use this to establish transects flown at altitudes typically 10 m to 30 m above ground; as low as safely possible".*

**9.1 Author Response:** We have made the change.

**10. Comment from Editor:** *Page 12, lines 1-2: "Increasing this threshold increases the link between*

*the calculated flux and a single land class, but reduces the number of footprints available for the analysis thus loosening the confidence interval". Would this be better as "Increasing this threshold increases the link between the calculated flux and a single land class, but reduces the number of footprints available for the analysis thus widening the confidence interval"?*

**10.1 Author Response:** We have made the change.

[revised manuscript text omitted]

---

## Author Response (AR3)

Response to Editor Comments:

**Editor Comment:** However, I am not sure if I agree with the conclusion drawn from data starvation test. Subsampling the original full 10 Hz time series at lower sampling frequencies should not lead to significant systematic flux underestimation, at least in the cases where the averaging period is much longer than the integral time scale of w's' (e.g. Lenschow et al., 1994; Bosveld and Beljaars, 2001; Rinne et al., 2008; Rinne and Ammann, 2012; Turnipseed et al., 2009). The data starvation actually simulates disjunct eddy sampling, rather than high frequency loss by inadequate response time. In order to properly estimate the effect of inadequate instrumental response time, the original time series should be filtered with low pass filter, rather than just subsampled. While you are probably right that the effect of high frequency loss on fluxes is relatively small, it should be shown in more rigorous way.

*Author Response:* We gave considerable thought to your suggestion that our analysis is similar to Disjunct Eddy Covariance (DEC) and came to the conclusion that as implemented is it not, as our analysis differs from DEC in its implicit filtering of the 10 $s^{-1}$ signal due to the interpolation. We could very likely implement it in a form similar to DEC. It would be interesting to see how that changes things, but it probably would not make a lot of difference in the computed flux. Responding to your call to treat the high-frequency loss in a more rigorous way we performed the test you suggested. The results turned out largely as you expected. The effect of high-frequency loss is 10% or less, and we have a more rigorous demonstration of that from the test. It turns out that the data starvation test we ran is quite analogous to the test you proposed. The difference is in the intentional use of a well-known filter in the explicit test versus some sort of implicit filter in the interpolation that we used for the data-starvation tests. We now have reported the results of both sets of tests. They give similar results, which we report in the revised manuscript. The result is now clearer in our heads, and hopefully also in the revised text. We appreciate your comments. They have indeed brought about improvements in the text. The comments of the original reviewers were instructive as well in that they challenged us to find some better ways to describe the Flux-Fragment Method without going into detail covered elsewhere.

*Manuscript Change:* In order to incorporate discussion of the tests as described above and to improve the readability, we have rewritten section 2.1 as follows:

[revised manuscript text omitted]